# AARL: Automated Auxiliary Loss for Reinforcement Learning

## Abstract

A good state representation is crucial to reinforcement learning (RL) while an ideal representation is hard to learn only with signals from the RL objective. Thus, many recent works manually design auxiliary losses to improve sample efficiency and decision performance. However, handcrafted auxiliary losses rely heavily on expert knowledge, and therefore lack scalability and can be suboptimal for boosting RL performance. In this work, we introduce Automated Auxiliary loss for Reinforcement Learning (AARL), a principled approach that automatically searches the optimal auxiliary loss function for RL. Specifically, based on the collected trajectory data, we define a general auxiliary loss space of size $4.6 \times 10^{19}$ and explore the space with an efficient evolutionary search strategy. We evaluate AARL on the DeepMind Control Suite and show that the searched auxiliary losses have significantly improved RL performance in both pixel-based and state-based settings, with the largest performance gain observed in the most challenging tasks. AARL greatly outperforms state-of-the-art methods and demonstrates strong generalization ability in unseen domains and tasks. We further conduct extensive studies to shed light on the effectiveness of auxiliary losses in RL.

## 1 Introduction

Reinforcement learning (RL) has been a hot research topic and has shown significant progress in many fields (Mnih et al., 2013; Silver et al., 2016; Gu et al., 2017; Vinyals et al., 2019). Recent RL research focuses on obtaining a good representation of states as it is shown to be the key to improve sample efficiency of RL (Laskin et al., 2020). This is because in high-dimensional environments, applying RL directly on the complex state inputs is incredibly sample-inefficient (Lake et al., 2017; Kaiser et al., 2019), making RL hard to scale to real-world tasks where interacting with the environment is costly (Dulac-Arnold et al., 2019). Standard RL paradigm learns the representation from critic loss (value prediction) and / or actor loss (maximizing cumulative reward), which hardly extracts informative representations in challenging environments like pixel-based RL and complex robotics systems (Lake et al., 2017; Kober et al., 2013).

This motivates adding auxiliary losses in support of learning better latent representations of states. The usage of auxiliary loss encodes prior knowledge of RL and its environments, and puts regularization on training (Shelhamer et al., 2016). Typical auxiliary losses are manually designed by human experts, including observation reconstruction (Yarats et al., 2019), reward prediction (Jaderberg et al., 2017) and environment dynamics prediction (Shelhamer et al., 2016; De Bruin et al., 2018; Ota et al., 2020). However, such auxiliary loss designs and choices rely heavily on human knowledge of what might be helpful for RL training, which requires extensive human efforts to scale to new tasks and can be suboptimal for improving RL performance (Yang & Nachum, 2021).

In this paper, we rigorously treat auxiliary loss functions for RL as a first-class problem to explicitly address the question: what is the universal approach to find a good auxiliary loss function for RL? Considering that the automated machine learning (AutoML) community has shown promising results with automated loss search in computer vision tasks (Li et al., 2019), we propose to automate the process of designing auxiliary loss functions of RL.

Specifically, we formulate the task as a bi-level optimization, where we try to find the best auxiliary loss function, which, to the most extent, helps train a good RL agent. The inner-level problem is a typical RL training problem, while the outer-level can be seen as a search problem. To tackle

Table 1: Existing auxiliary losses in our search space.

| Auxiliary loss | Loss operator | Loss inputs | | |
|---|---|---|---|---|
| | | Horizon | Source | Target |
| Forward dynamics | MSE | 1 | $\{s_t, a_t\}$ | $\{s_{t+1}\}$ |
| Inverse dynamics | MSE | 1 | $\{a_t, s_{t+1}\}$ | $\{s_t\}$ |
| Reward prediction | MSE | 1 | $\{s_t, a_t\}$ | $\{r_t\}$ |
| Action inference | MSE | 1 | $\{s_t, s_{t+1}\}$ | $\{a_t\}$ |
| CURL (Laskin et al., 2020) | Bilinear | 1 | $\{s_t\}$ | $\{s_t\}$ |
| ATC (Stooke et al., 2021) | Bilinear | k | $\{s_t\}$ | $\{s_{t+1}, \cdots, s_{t+k}\}$ |
| SPR (Schwarzer et al., 2020) | N-MSE | k | $\{s_t, a_t, a_{t+1}, \cdots, a_{t+k-1}\}$ | $\{s_{t+1}, \cdots, s_{t+k}\}$ |

this, we first design a novel search space of auxiliary loss functions, which is a combination of two components: loss input and loss operator. As shown in Table 1, the search space covers many existing handcrafted losses, and it is also substantially large (as large as $4.6 \times 10^{19}$). We then propose an efficient search strategy to explore the space. This is done in a two-step manner. We first finalize the loss operator specification and then use an evolutionary strategy that performs mutations on configurations of loss input, to identify the top-performing loss inputs quickly.

We evaluate our search framework on both pixel-based and state-based environments in the Deep-Mind Control suite (Tassa et al., 2018). Extensive experiments show that the searched auxiliary loss functions greatly outperform state-of-the-art methods and can easily transfer to unseen environments (that are never used during the search), showing that our proposed method is robust and effective.

We highlight the main contributions of this paper below:

- We introduce a principled and universal approach for auxiliary loss design in RL. To the best of our knowledge, we are the first to derive the optimal auxiliary loss function with an automatic process. Our framework can be easily applied to arbitrary RL tasks to search for the best auxiliary loss function.

- We demonstrate that AARL significantly outperforms state-of-the-art methods in both pixel-based and state-based environments. The searched auxiliary loss functions show strong generalization ability to transfer to unseen environments.

- We analyze the derived auxiliary loss functions and deliver some insightful discussions that we hope will deepen the understanding of auxiliary losses in RL. We will also open source our codebase to facilitate future research.

## 2 METHODOLOGY

We consider the standard Markov Decision Process (MDP) setting where we specify the state, action and reward at time step $t$ as $(s_t, a_t, r_t)$. The sequence of interaction data is denoted as $(s_t, a_t, r_t, \cdots, s_{t+k})$, where $k$ represents the *horizon* of this sequence. Suppose we have an RL agent $M_\omega$ parameterized by $\omega$ and $\mathcal{R}(M_\omega; \mathcal{E})$ is the agent performance (i.e., cumulative discounted reward) in environment $\mathcal{E}$. The goal of AARL is to find the optimal auxiliary loss function $\mathcal{L}$, such that, when $\omega$ is optimized under an arbitrary RL objective function $\mathcal{L}_{\text{RL}}$ (e.g., actor-critic loss) together with $\mathcal{L}$, the agent achieves the best performance. Formally, we have

$$
\begin{aligned}
\max_{\mathcal{L}} \quad & \mathcal{R}(M_{\omega^*(\mathcal{L})}; \mathcal{E}), \\
\text{s.t.} \quad & \omega^*(\mathcal{L}) = \arg\min_{\omega}(\mathcal{L}_{\text{RL}}(\omega; \mathcal{E}) + \lambda\mathcal{L}(\omega; \mathcal{E})),
\end{aligned}
\tag{1}
$$

where $\lambda$ is a hyper-parameter that controls the relative weight of the auxiliary task. We solve this bi-level optimization problem with AutoML techniques. The inner optimization is a standard RL training procedure. For the outer one, we define a finite and discrete search space (Section 2.1), and use a variation of evolution strategy to explore the space (Section 2.2). We will explain the details in the rest of this section.

### 2.1 SEARCH SPACE

An auxiliary loss function $\mathcal{L}$ can be viewed as a combination of two components: 1) loss input $\mathcal{I}$ and 2) loss operator $f$. We define a search space as shown in Figure 1, where $\mathcal{I}$ is a pair of binary masks

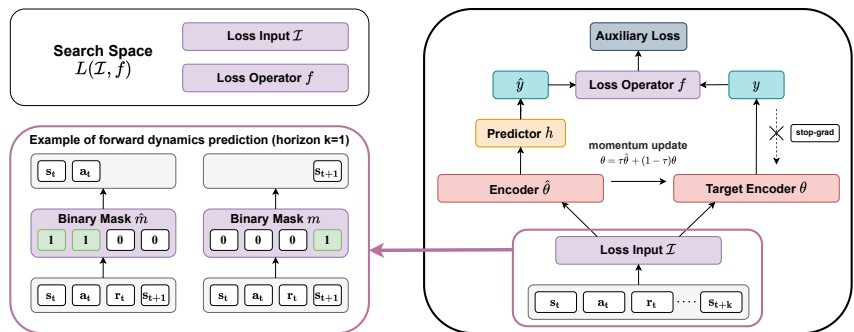

Figure 1: Computation graph and search space of auxiliary loss functions.

$(\hat{m}, m)$ that selects from interaction data to compute $\hat{y}$ and $y$, and $f$ is an operator that aggregates the selected inputs into an expression of the loss function with scalar output.

**Loss Input**  Unlike supervised machine learning tasks with an explicit prediction and ground truth for the loss function, auxiliary losses in RL have no ground truths beforehand. Instead, they are generated accordingly upon interaction with the environment itself in a self-supervised manner. As shown in Figure 1, we take some tokens from the sequence $(s_t, a_t, r_t, \cdots, s_{t+k})$ with binary mask vector $\hat{m}$ and feed them into the encoder $\hat{\theta}$ which maps states to latent representations. The predictor module then tries to predict *targets* (encoded with a momentum encoder $\theta$, as is typically done in recent works), which are another few tokens we select from the sequence with another binary mask vector $m$. Details about momentum encoder are given in Appendix A.1.2. In other words, the search space for loss input is a pair of binary masks $(\hat{m}, m)$, each of which is up to length $(3k + 1)$ if the length of an interaction data sequence, i.e., horizon, is limited to $k$ steps. In our case, we set the maximum horizon length $k_{\max} = 10$.

**Loss Operator**  We optimize $\hat{y}$ to be as similar as possible to $y$. Therefore, we make loss operator $f$ cover commonly-used similarity measures, including inner product (Inner) (He et al., 2020; Stooke et al., 2021), bilinear inner product (Bilinear) (Laskin et al., 2020), cosine similarity (Cosine) (Chen et al., 2020), mean squared error (MSE) (Ota et al., 2020; De Bruin et al., 2018) and normalized mean squared error (N-MSE) (Schwarzer et al., 2020). Some existing works, e.g., contrastive objectives like InfoNCE loss (Oord et al., 2018), also incorporate the trick to sample un-paired predictions and targets as negative samples and maximize the distances between them. We find this technique applicable to all the loss operators mentioned above and thus incorporate the discriminative version of these operators in our search space.

**Search Space Complexity**  As shown in Table 1, many existing manually designed auxiliary losses can naturally fit into our loss space thus are our special cases, which proves that this loss space is reasonable, flexible, and general. However, this is also at the cost that the space is substantially large. In total, the size of the entire space is $10 \times \sum_{i=1}^{10} 2^{6i+2} \approx 4.6 \times 10^{19}$. The detailed derivation can be found in Appendix C.

## 2.2 SEARCH STRATEGY

**Search Space Pruning**  Considering that the loss space is extremely large, an effective optimization strategy is inevitably required. Directly grid-searching over the whole space is infeasible because of unacceptable computational cost. Thus some advanced techniques such as space pruning and an elaborate search strategy are necessary. Our search space can be seen as a combination of the space for the input $\mathcal{I}$ and the space for the operator $f$. Inspired by AutoML works (Dai et al., 2020; Ying et al., 2019) that search for hyper-parameters first and then neural architectures, we approximate the joint search of input and operator in Equation (1) in a two-step manner. The optimal

Table 2: Normalized episodic rewards (mean & standard deviation for 5 seeds) of 3 environments used in evolution on pixel-based DMControl500K with different loss operators.

| Loss operator and discrimination | Inner | Bilinear | Cosine | MSE | N-MSE |
|---|---|---|---|---|---|
| w/ negative samples | $0.979 \pm 0.344$ | $0.953 \pm 0.329$ | $0.872 \pm 0.412$ | $0.124 \pm 0.125$ | $0.933 \pm 0.360$ |
| w/o negative samples | $0.669 \pm 0.311$ | $0.707 \pm 0.299$ | $0.959 \pm 0.225$ | $\mathbf{1.000 \pm 0.223}$ | $0.993 \pm 0.229$ |

auxiliary loss $\{\mathcal{I}^*, f^*\}$ can be optimized as:

$$\max_{\mathcal{L}} \mathcal{R}(M_{\omega^*(\mathcal{L})}; \mathcal{E}) = \max_{\mathcal{I},f} \mathcal{R}(M_{\omega^*(\mathcal{I},f)}; \mathcal{E}) \approx \max_{\mathcal{I}} \mathcal{R}(M_{\omega^*(\mathcal{I},f^*)}; \mathcal{E})$$

$$\text{where} \quad f^* \approx \arg\max_{f} \mathbb{E}_{\mathcal{I}}[\mathcal{R}(M_{\omega^*(\mathcal{I},f)}; \mathcal{E})] \tag{2}$$

To decide the best loss operator, for every $f$ in the loss operator space, we estimate $\mathbb{E}_{\mathcal{I}}[\mathcal{R}(M_{\omega^*(\mathcal{I},f)}; \mathcal{E})]$ with a random sampling strategy. We run 15 trials for each loss operator and calculate the average of their normalized return on three environments. More details are available in Appendix A.1.3. Surprisingly, as summarized in Table 2, the simplest MSE without negative samples outperforms all other loss operators with complex designs. Therefore, this loss operator is chosen for the rest of this paper.

**Evolution** Even with the search space pruning, the rest of the space still remains large. To efficiently identify top candidates from the search space, we adopt an evolutionary algorithm (Real et al., 2019). The pipeline of the evolution process is illustrated in Figure 2. In each stage, we first train and evaluate a population of candidates with size $P = 100$, where each candidate corresponds to a pair of binary masks $(\hat{m}, m)$. The candidates are then sorted by the approximated *area under learning curve* (AULC) (Ghiassian et al., 2020; Stadie et al., 2015), which well captures both convergence speed and final performance (Viering & Loog, 2021) and reduces the variance of RL evaluation.

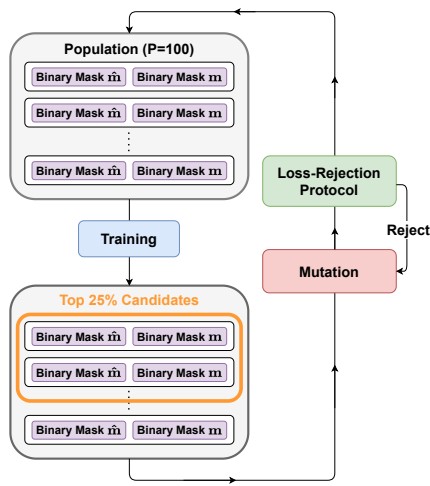

Figure 2: Overview of the evolution pipeline.

The top-25% candidates are selected after each training stage, and we perform four types of mutations on the selected candidates to form a new population for the next stage: replacement (50% of the population); crossover (20%); horizon decrease and horizon increase (10%), as shown in Figure 3. Moreover, the last 20% of the new population is from random generation. As for each bit of masks, replacement operation flips the given bit with probability $p = \frac{1}{2 \cdot (3k+1)}$, where $k$ is the horizon length. Crossover generates a new candidate by randomly combining the mask bits of two candidates with the same horizon length in the population. We also incorporate the knowledge from the existing auxiliary loss designs by bootstrapping the initial population with a prior distribution (Co-Reyes et al., 2020), so that we can quickly converge to a reasonable loss function. More implementation details are available in Appendix A.3.

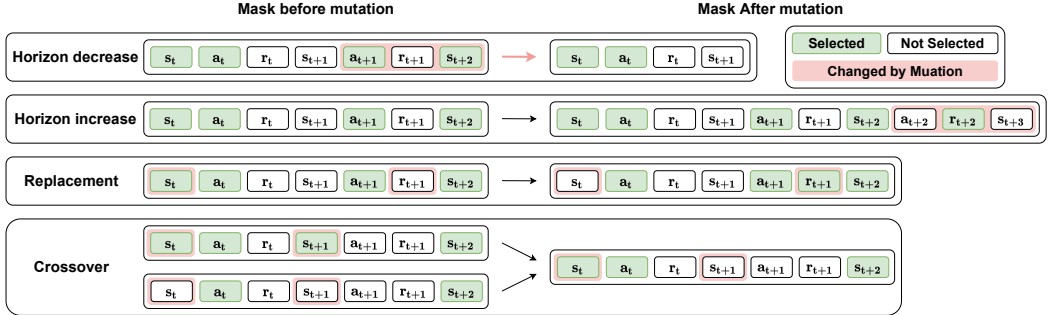

Figure 3: Mutation operations of auxiliary loss functions.

# 3 EXPERIMENTS

We evaluate our method and the obtained auxiliary loss functions on DMControl suite (Tassa et al., 2018). DMControl suite is powered by MuJoCo physics engine (Todorov et al., 2012) with various challenging tasks. In both pixel-based (images) and state-based (proprioceptive features) settings, we choose Soft Actor-Critic (SAC) (Haarnoja et al., 2018) as our base RL algorithm since it is the state-of-the-art model-free RL algorithm. Implementation details are given in Appendix A.

## 3.1 PIXEL-BASED RL

**Experiment Settings**   We follow the same network architecture as CURL with a 4-layer convolutional encoder for image input. In the search phase, we compare our method AARL to SAC, SAC (no aug), and CURL with the same hyper-parameters reported in Appendix A.2.1. All methods randomly crop images from $100 \times 100$ to $84 \times 84$ as data augmentation apart from SAC (no aug). After searching, we test the generalization ability of the obtained auxiliary losses in other unseen environments in comparison with state-of-the-art model-free and model-based methods.

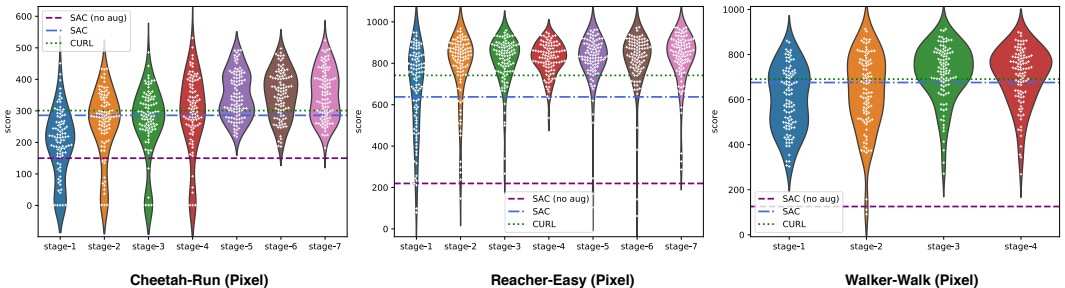

Figure 4: Evolution process in pixel-based environments. Every white dot represents a loss candidate, and the score of y-axis shows its corresponding approximated AULC score (Ghiassian et al., 2020; Stadie et al., 2015). The horizontal lines show the scores of baselines. The AULC score is approximated with the average evaluation score at 100k, 200k, 300k, 400k, 500k time steps.

**Search Results**   We apply our search algorithm on Cheetah-Run, Reacher-Easy, and Walker-Walk, which we call "*training environments*". We choose these environments because they are more challenging and there might be more room for improvement with auxiliary losses. For each environment, we set the total budget of each experiment to 16k GPU hours (on NVIDIA P100) and terminate the experiment when the resource is exhausted. The evolution process is demonstrated in Figure 4, where the y-axis shows the AULC of candidates. The results show that, during the evolution process, we can easily find candidates surpassing baselines in the first stage. In the subsequent stages, the overall population score continues to grow, and most candidates can outperform baselines. This demonstrates the effectiveness of our search space and the efficiency of our search strategy.

**Generalize to Other Environments**   To reduce the risk that auxiliary losses overfit to one particular environment and avoid selecting candidates that perform well only due to randomness, we run cross-validation on the three searched environments to decide the best candidate. The details of this process can be found in the Appendix A.4. After cross-validation, we finalize an auxiliary loss function, which we call "AARL-Pixel" (all the top candidates during evolution are reported in Appendix D), and we train agents with the obtained auxiliary loss to compare with state-of-the-art model-free and model-based methods on DMControl100k and DMControl500k. We select these environments because they are common benchmarks for state-of-the-art algorithms of pixel-based RL. (Laskin et al., 2020). Note these comparisons are additionally made on three unseen environments, that are entirely not accessible during evolution, to test the generalizability of our method and to ensure our comparisons are fair. The results are summarized in Table 3, where AARL greatly outperforms state-of-the-art methods on 11 out of 12 benchmark settings. Note that Finger-Spin, Reacher-Easy and Ball in cup-Catch are environments that are never used during the search, while AARL still performs very well on them. This result implies that AARL-Pixel is a robust and effective auxiliary loss that is potentially helpful to RL under various settings. The performance gain

Table 3: Episodic rewards (mean & standard deviation for 10 seeds) on DMControl100K and DM-Control500K with pixel inputs. Note that the optimal score of DMControl is 1000 for all environments. The baselines methods are PlaNet (Hafner et al., 2019b), Dreamer (Hafner et al., 2019a), SAC+AE (Yarats et al., 2019), SLAC (Lee et al., 2019), pixel-based SAC (Haarnoja et al., 2018). Performance values of all baselines are referred to Laskin et al. (2020), except for Pixel SAC. We also show learning curves of all 12 DMC environments in Appendix B.1.

| 500K Steps Scores | AARL-Pixel | CURL§ | PlaNet§ | Dreamer§ | SAC+AE§ | SLACv1§ | Pixel SAC |
|---|---|---|---|---|---|---|---|
| Finger-Spin* | **983 ± 4** | 926 ± 45 | 561 ± 284 | 796 ± 183 | 884 ± 128 | 673 ± 92 | 282 ± 102 |
| Cartpole-Swingup* | **864 ± 19** | 841 ± 45 | 475 ± 71 | 762 ± 27 | 735 ± 63 | - | 344 ± 104 |
| Reacher-Easy† | **938 ± 46** | 929 ± 44 | 210 ± 390 | 793 ± 164 | 627 ± 58 | - | 312 ± 132 |
| Cheetah-Run† | 613 ± 39 | 518 ± 28 | 305 ± 131 | 570 ± 253 | 550 ± 34 | **640 ± 19** | 99 ± 28 |
| Walker-Walk† | **917 ± 18** | 902 ± 43 | 351 ± 58 | 897 ± 49 | 847 ± 48 | 842 ± 51 | 76 ± 44 |
| Ball in cup-Catch* | **970 ± 8** | 959 ± 27 | 460 ± 380 | 897 ± 87 | 794 ± 58 | 852 ± 71 | 200 ± 114 |
| 100K Steps Scores | | | | | | | |
| Finger-Spin* | **872 ± 27** | 767 ± 56 | 136 ± 216 | 341 ± 70 | 740 ± 64 | 693 ± 141 | 160 ± 138 |
| Cartpole-Swingup* | **815 ± 66** | 582 ± 146 | 297 ± 39 | 326 ± 27 | 311 ± 11 | - | 243 ± 19 |
| Reacher-Easy† | **778 ± 164** | 538 ± 223 | 20 ± 50 | 314 ± 155 | 274 ± 14 | - | 277 ± 69 |
| Cheetah-Run† | **449 ± 34** | 299 ± 48 | 138 ± 88 | 235 ± 137 | 267 ± 24 | 319 ± 56 | 128 ± 12 |
| Walker-Walk† | **510 ± 151** | 403 ± 24 | 224 ± 48 | 277 ± 12 | 394 ± 22 | 361 ± 73 | 127 ± 28 |
| Ball in cup-Catch* | **862 ± 167** | 769 ± 43 | 0 ± 0 | 246 ± 174 | 391 ± 82 | 512 ± 110 | 100 ± 90 |

†: Training environments. *: Unseen environments. §: Results reported in Laskin et al. (2020).

is obvious in DMControl100K, where AARL-Pixel (pixel-based input) has shown great sample efficiency.

## 3.2 STATE-BASED RL

**Experiment Settings** As for state-based RL, since the state is low-dimensional, we simply use a 1-layer densely connected MLP as the state encoder as shown in Figure 5. So, for this setting, we focus on this simple encoder structure. Additional ablations on state encoder architectures are given in Section 3.4. In the search phase, we compare AARL to SAC-Identity, SAC-DenseMLP, CURL-DenseMLP. To ensure a fair comparison, all SAC related hyper-parameters are the same as those reported in the CURL paper. Details can be found in Appendix A.2.2. SAC-Identity is vanilla SAC with no state encoder, while the other three methods (AARL, SAC-DenseMLP, CURL-DenseMLP) use the same encoder architecture. Different from the pixel-based setting, there is no data augmentation in the state-based setting. Note that many environments that are challenging in pixel-based settings become easy to tackle with state-based inputs. Therefore we apply our search framework to more challenging environments for state-based RL, including Cheetah-Run, Hopper-Hop and Quadruped-Run.

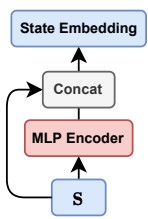

Figure 5: Network architecture of 1-layer DenseMLP state encoder.

**Search Results** Similar to pixel-based settings, we approximate AULC with the average score agents achieve at 300k, 600k, 900k, 1200k and 1500k time steps[1]. For each environment, we early stop the experiment when the budget of 1,500 GPU hours is exhausted. The evolution process is shown in Figure 6, where we find a large portion of candidates outperform baselines (horizontal dashed lines). The performance improvement is especially significant on Cheetah-Run, where almost all candidates in the population greatly outperform all baselines by the end of the first stage.

**Generalize to Other Environments** Similar to pixel-based settings, we also use cross-validation to select the best loss function, which we call "AARL-State" here (all the top candidates during evolution are reported in Appendix D), and run a thorough evaluation on all 18 environments. The results are summarized in Table 4. AARL-State again brings significant performance gain, achieving much stronger sample efficiency than SAC. It is noteworthy that AARL-State is able to outperform baseline methods in 16 out of 18 environments (with 15 unseen environments). These results show that even for tasks with lower-dimensional state space, there is still a huge potential for improvement with better auxiliary objectives. Moreover, this performance gain is especially significant

---

[1] As for Cheetah-Run, we still use average score agents achieve at 100k, 200k, 300k, 400K and 500k time steps since agents converge close to optimal score within 500k time steps.

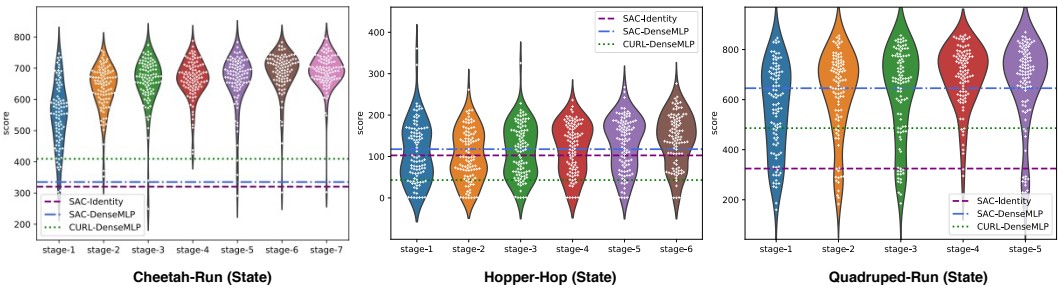

Figure 6: Evolution process in state-based environments. Every white dot represents a loss candidate, and the score of y-axis shows its corresponding approximated AULC score. The horizontal lines show the scores of baselines. The AULC score is approximated with the average evaluation score at 300k, 600k, 900k, 1200k, 1500k time steps (Cheetah-Run at 100k, 200k, 300k, 400K).

Table 4: Episodic rewards (mean & standard deviation for 10 seeds) on DMControl100K (easy tasks) and DMControl1000K (difficult tasks) with state inputs. SAC-Identity has no state encoder while AARL, SAC and CURL use the same state encoder architecture. All four variants here use the same hyper-parameters.

| 100K Steps Scores | AARL-State | SAC-Identity | SAC | CURL |
|---|---|---|---|---|
| Finger-Spin* | **837 ± 52** | 805 ± 32 | 785 ± 106 | 712 ± 83 |
| Finger-Turn hard* | 218 ± 117 | **347 ± 150** | 174 ± 94 | 43 ± 42 |
| Cartpole-Swingup* | **877 ± 5** | 873 ± 10 | 866 ± 7 | 854 ± 17 |
| Cartpole-Swingup sparse* | **695 ± 147** | 455 ± 359 | 627 ± 307 | 446 ± 196 |
| Reacher-Easy* | **934 ± 38** | 697 ± 192 | 874 ± 87 | 749 ± 183 |
| Cheetah-Run† | **472 ± 30** | 237 ± 27 | 172 ± 29 | 190 ± 32 |
| Walker-Stand* | **948 ± 7** | 940 ± 10 | 862 ± 196 | 767 ± 104 |
| Walker-Walk* | 906 ± 78 | 873 ± 89 | **925 ± 22** | 852 ± 64 |
| Walker-Run* | **564 ± 45** | 559 ± 34 | 403 ± 43 | 289 ± 61 |
| Ball in cup-Catch* | **965 ± 7** | 954 ± 12 | 962 ± 13 | 941 ± 32 |
| Fish-Upright* | **498 ± 88** | 471 ± 62 | 400 ± 62 | 295 ± 117 |
| Hopper-Stand* | **311 ± 177** | 14 ± 16 | 26 ± 40 | 6 ± 3 |

†: Training environments. ∗: Unseen environments.

| 1000K Steps Scores | AARL-State | SAC-Identity | SAC | CURL |
|---|---|---|---|---|
| Quadruped-Run† | **838 ± 58** | 345 ± 157 | 707 ± 148 | 497 ± 128 |
| Pendulum-Swingup* | **579 ± 410** | 506 ± 374 | 379 ± 391 | 363 ± 366 |
| Hopper-Hop† | **278 ± 106** | 121 ± 51 | 134 ± 93 | 60 ± 22 |
| Humanoid-Stand* | **286 ± 15** | 9 ± 2 | 7 ± 1 | 7 ± 1 |
| Humanoid-Walk* | **299 ± 55** | 16 ± 28 | 2 ± 0 | 2 ± 0 |
| Humanoid-Run* | **88 ± 2** | 1 ± 0 | 1 ± 0 | 1 ± 0 |

†: Training environments. ∗: Unseen environments.

in complex environments like Humanoid, where SAC barely learns anything at 1000K time steps, while AARL-State is able to achieve much better performance.

## 3.3 Analysis of Auxiliary Loss Functions

We have collected a large number of data from the evolution process so far. To make the best use of these data, we now conduct a comprehensive analysis to investigate statistical relations between patterns of auxiliary loss functions and RL performance.

**Typical Patterns** We first analyze how RL performance is affected by typical patterns of auxiliary loss functions. Define the set of the input to the encoder that is being trained as "source", and the set of prediction targets as "target". We say an auxiliary loss candidate has a certain pattern if the pattern's source is a subset of the candidate's source, and the pattern's target is a subset of the candidate's target. For instance, a loss candidate of $\{s_t, a_t\} \rightarrow \{s_{t+1}, s_{t+2}\}$ has the pattern $\{s_t, a_t\} \rightarrow \{s_{t+1}\}$, and does not have the pattern $\{a_t, s_{t+1}\} \rightarrow \{s_t\}$. The patterns we consider include forward dynamics $\{s_t, a_t\} \rightarrow \{s_{t+1}\}$, inverse dynamics $\{a_t, s_{t+1}\} \rightarrow \{s_t\}$, reward prediction $\{s_t, a_t\} \rightarrow \{r_t\}$, action inference $\{s_t, s_{t+1}\} \rightarrow \{a_t\}$ and state reconstruction in the latent space $\{s_t\} \rightarrow \{s_t\}$. For each of these patterns, we divide auxiliary loss candidates into two classes: 1) with this pattern or 2) without this pattern. We then calculate the average performance of these two classes and compare their performances to conclude whether this pattern is helpful for RL. The results are summarized in Table 5. Here a positive number indicates that this pattern is beneficial, and if the performance gain is statistically significant, the number is marked with the asterisk, indicating it is very likely to be helpful. A negative number indicates that this pattern is detrimental and correlated to worse performance. We highlight some interesting observations as follows. 1) Forward dynamics is helpful in most environments, and improves the RL performance on Reacher-Easy (Pixel) and Cheetah-Run (State) significantly (i.e., p-value<0.05). 2) State reconstruction in the latent space is able to improve RL performance in pixel-based environments while it has significant negative effects on state-based environments. We hypothesize that this is because in the pixel-based setting, data augmentation encourages the encoder to learn augmentation-invariant representations, while in the state-based setting, no augmentation is used, and thus the encoder learns no useful rep-

resentations. This also explains why CURL is underperforming in state-based experiments. Note that not all typical patterns bring performance gains. Some can even be very detrimental, and this shows that commonly-used auxiliary loss patterns do not work well in the state-based setting, further highlighting the fact that great research potential lies in this setting.

Table 5: Statistical analysis on auxiliary loss functions. The number reported is the difference of the mean score of two classes of auxiliary losses during evolution, and we report its corresponding p-value from the t-test.

| | Typical patterns (w/ - w/o) | | | | |
|---|---|---|---|---|---|
| | Forward dynamics | Inverse dynamics | Reward prediction | Action inference | State reconstruction |
| Cheetah-Run (Pixel) | $+1.28$ | $-3.51$ | $-31.16^{**}$ | $-75.95^{**}$ | $+42.44^{**}$ |
| Reacher-Easy (Pixel) | $+28.25^{*}$ | $+8.36$ | $+37.80^{**}$ | $+3.35$ | $+70.72^{**}$ |
| Walker-Walk (Pixel) | $+22.20$ | $-48.59^{**}$ | $-8.11$ | $+29.86^{*}$ | $+13.93$ |
| Cheetah-Run (State) | $+94.18^{**}$ | $-23.66^{**}$ | $-33.28^{**}$ | $-109.33^{**}$ | $-50.15^{**}$ |
| Hopper-Hop (State) | $+15.50^{**}$ | $-16.47^{**}$ | $-11.30^{*}$ | $-32.10^{**}$ | $-25.67^{**}$ |
| Quadruped-Run (State) | $-28.07$ | $-18.19$ | $-114.23^{**}$ | $-105.37^{**}$ | $-82.06^{**}$ |

∗: p-value $< 0.05$. ∗∗: p-value $< 0.01$

| | Source or target | | |
|---|---|---|---|
| | State, $n_{target} > n_{source}$ | Action, $n_{target} > n_{source}$ | Reward, $n_{target} > n_{source}$ |
| Cheetah-Run (Pixel) | $+80.09^{**}$ | $+13.62$ | $+3.33$ |
| Reacher-Easy (Pixel) | $+1.98$ | $-12.72$ | $+65.66^{**}$ |
| Walker-Walk (Pixel) | $+73.56^{**}$ | $+42.22^{*}$ | $-41.90^{*}$ |
| Cheetah-Run (State) | $+188.06^{**}$ | $-102.62^{**}$ | $-93.94^{**}$ |
| Hopper-Hop (State) | $+19.80^{**}$ | $-29.70^{**}$ | $-5.03$ |
| Quadruped-Run (State) | $+75.17^{**}$ | $-4.31$ | $-46.60^{*}$ |

∗: p-value $< 0.05$. ∗∗: p-value $< 0.01$

**Number of Sources and Targets**  We then investigate whether it is more beneficial to use a small number of sources to predict a large number of targets ($n_{target} > n_{source}$, e.g., use $s_t$ to predict $s_{t+1}, s_{t+2}, s_{t+3}$), or the other way around ($n_{target} < n_{source}$, e.g., use $s_t, s_{t+1}, s_{t+2}$ to predict $s_{t+3}$). The statistical results are shown in Table 5, where we find that auxiliary losses with more states on the target side have a significant advantage over losses with more states on the source side. This result echoes recent works (Stooke et al., 2021; Schwarzer et al., 2020) which show that predicting more states leads to strong performance gains.

### 3.4   ADDITIONAL ABLATION STUDY

**Search Space Pruning**  As introduced in Section 2.2, we decompose the full search space into loss operator and loss inputs. Here we try to directly apply evolution strategy to the whole space without the pruning step. The comparison results are shown in Figure 7. We can see that pruning improves the evolution process, making it easier to find good candidates.

**Encoder Architecture for State-based RL**  As shown in Figure 5, we choose a 1-layer densely connected MLP as the state encoder for state-based RL. Here, we conduct an ablation study on different encoder architectures in the state-based setting. The results are summarized in Table 6, where AARL with 4-layer encoders consistently perform worse than 1-layer encoders. We also note that dense connection is helpful in the state-based setting compared with naive MLP encoders.

Table 6: Normalized episodic rewards of AARL (mean & standard deviation for 5 seeds of 6 environments) on state-based DMControl100K with different encoder architectures.

| AARL-MLP (1-layer) | AARL-MLP (4-layer) |
|---|---|
| $0.919 \pm 0.217$ | $0.544 \pm 0.360$ |
| AARL-DenseMLP (1-layer) | AARL-DenseMLP (4-layer) |
| $\mathbf{1.000 \pm 0.129}$ | $0.813 \pm 0.218$ |

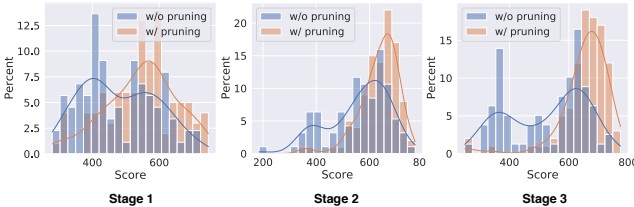

Figure 7: Comparison of evolution with and without pruning by performance histogram.

## 4 RELATED WORK

### 4.1 REINFORCEMENT LEARNING WITH AUXILIARY LOSSES

Using auxiliary tasks to improve sample efficiency of RL, especially on pixel-based control tasks, has been explored in many recent works. A number of manually designed auxiliary objectives are shown to boost RL performance, including observation reconstruction (Yarats et al., 2019), reward prediction (Jaderberg et al., 2017), dynamics prediction (De Bruin et al., 2018) and contrastive learning objectives (Laskin et al., 2020; Schwarzer et al., 2020; Stooke et al., 2021). It is noteworthy that these works mainly focus on pixel-based settings, while only a limited number of works study the state-based setting (Munk et al., 2016; Ota et al., 2020). Although it might seem that the state-based setting benefits less from auxiliary tasks due to their lower-dimensional state space, we show that there is in fact a huge potential of improving state-based RL performance with auxiliary objectives.

Compared to the previous works, we point out two major advantages of our approach. 1) Instead of handcrafting an auxiliary loss with expert knowledge, AARL automatically searches for the best auxiliary loss, relieving researchers from such tedious work. 2) AARL is a principled approach that can be used in arbitrary RL settings. In both pixel-based and the rarely studied state-based settings, we discover good auxiliary losses that bring significant performance improvement.

### 4.2 AUTOMATED REINFORCEMENT LEARNING

It is well known that RL training is sensitive to hyper-parameters and environment changes (Henderson et al., 2018). Thus many works have attempted to use techniques in AutoML to alleviate human intervention. Exiting works focus on hyper-parameter optimization (Espeholt et al., 2018; Paul et al., 2019; Xu et al., 2020; Zahavy et al., 2020), reward search (Faust et al., 2019; Veeriah et al., 2019) and network architecture search (Runge et al., 2019; Franke et al., 2021). In contrast, our method aims to search for auxiliary loss functions that generalize across different environments.

### 4.3 AUTOMATED LOSS DESIGN

A few recent works in the AutoML community have been automating the design of good loss functions that outperform traditionally handcrafted ones. Specifically, AM-LFS (Li et al., 2019) defines the loss function search space as a parameterized probability distribution of softmax loss hyperparameters. AutoLoss-Zero (Li et al., 2021) proposes to search loss functions with primitive mathematical operators. These methodologies are proposed specifically for computer vision tasks.

For RL, existing works focus on searching for a better RL objective, EPG (Houthooft et al., 2018) and MetaGenRL (Kirsch et al., 2020) define the search space of loss functions as parameters of a low complexity neural network. Co-Reyes et al. (2020) defines the search space of RL loss functions as a directed acyclic graph and discovers two DQN-like regularized RL losses. Note that none of these works investigate auxiliary loss functions, which are crucial to facilitate representation learning in RL and to make RL successful in highly complex environments. To the best of our knowledge, our work is the first attempt to search for auxiliary loss functions and greatly improve RL performance.

## 5 CONCLUSION AND FUTURE WORK

We present AARL, a principled and universal approach for automated auxiliary loss design for RL. With this framework, we discover highly performant auxiliary loss functions that generalize beyond the training environments for both pixel-based and state-based settings. We present extensive experimental results that provide strong empirical evidence for the effectiveness of our method, and also conduct an in-depth investigation of the statistical relations between auxiliary loss patterns and RL performance. We hope our studies provide insights that will deepen the understanding of auxiliary losses in RL, and shed light on how to make RL more efficient and practical. One interesting future work direction is to further investigate the relationship between auxiliary loss patterns and RL performance, and to better understand how and why auxiliary losses can help to improve RL performance. Another direction is to study how to combine the RL objective and the auxiliary loss functions in a more effective manner.

## REPRODUCIBILITY STATEMENT

All implementation details are introduced in the main text and appendix (Appendix A). We will open source our codebase to facilitate future research.

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

# A  IMPLEMENTATION DETAILS

## A.1  ARCHITECTURE AND SEARCH SPACE

### A.1.1  STATE ENCODER ARCHITECTURES

We demonstrate in Figure 8 the overall architecture where our auxiliary loss is used. The architecture is generally identical to those adopted in CURL (Laskin et al., 2020). "Pixel-based" and "State-based" are the architectures we used in our experiments. "MLP" and "4-layer DenseMLP" are for ablations.

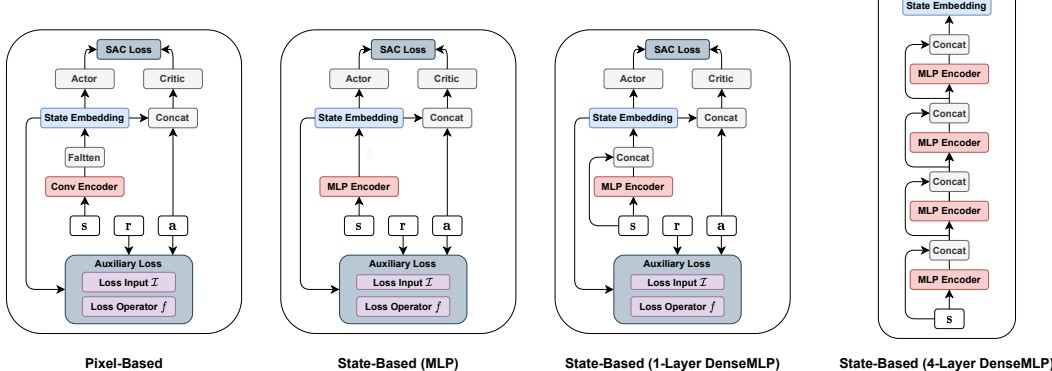

Figure 8: Network structures of pixel-based RL and state-based RL with auxiliary losses.

### A.1.2  SIAMESE NETWORK

For a fair comparison with baseline methods, we follow the same Siamese networks structure for representation learning as CURL (Laskin et al., 2020). As shown in Figure 1, when computing targets $y$ for auxiliary losses, we map states to state embeddings with a target encoder. We stop gradients from target encoder $\theta$ and update $\theta$ in the exponential moving averaged (EMA) manner where $\theta' = \tau\hat{\theta} + (1 - \tau)\theta$. This step, i.e., to freeze the gradients of the target encoder, is necessary when the loss is computed without negative samples. Otherwise, encoders will collapse to generating the same representation for any input. We have verified this in our early experiments.

### A.1.3  LOSS OPERATORS

**Instance Discrimination**  Our implementation is based on InfoNCE loss (Oord et al., 2018):

$$L = \log \frac{\exp(\phi(\hat{y}, y_+))}{\exp(\phi(\hat{y}, y_+)) + \sum_{i=0}^{K-1} \exp(\phi(\hat{y}, y_i))} \tag{3}$$

The instance discrimination loss can be interpreted as the log-loss of a K-way softmax classifier whose label is $y_+$. The difference between discrimination based loss operators lies in the discrimination objective $\phi$ used to measure agreement between $(\hat{y}, y)$ pairs. Inner employs inner product $\phi(\hat{y}, y) = \hat{y}^\top y$ while Bilinear employs bilinear product $\phi(\hat{y}, y) = \hat{y}Wy$, where $W$ is a learnable parameter matrix. Cosine uses cosine distance $\phi(\hat{y}, y) = \frac{\hat{y}^\top y}{\|\hat{y}\| \cdot \|y\|}$ for further matrix calculation. As for cross entropy based loss without negative samples, we only take diagonal elements for matrix $M$ where $M_{i,j} = \phi(\hat{y}_i, y_j)$ for cross entropy calculation.

**Mean Squared Error**  The implementation of MSE-based loss operators are straightforward. MSE loss operator = $(\hat{y} - y_+)^2$ while normalized MSE = $(\frac{\hat{y}}{\|\hat{y}\|} - \frac{y_+}{\|y_+\|})^2$. When combined with negative samples, MSE loss operator (with negative pairs) = $(\hat{y} - y_+)^2 - (\hat{y} - y_-)^2$ while normalized MSE (with negative pairs) = $(\frac{\hat{y}}{\|\hat{y}\|} - \frac{y_+}{\|y_+\|})^2 - (\frac{\hat{y}}{\|\hat{y}\|} - \frac{y_-}{\|y_-\|})^2$.

**Evaluation of Performance Expectation**   To reduce the large search space consists of loss input $\mathcal{I}$ and loss operator $f$, we run 15 trials for each loss operator to get an estimation of performance expectation. For each of 10 possible $f$ in the search space (5 operators with optional negative samples), we run 5 trials on each of the 3 pixel-based environments (used in evolution) with the same loss inputs $\{s_t, a_t\} \rightarrow \{s_{t+1}\}$, as we found that forward dynamics is a reasonable representative of our search space with highly competitive performance.

## A.2   TRAINING DETAILS

### A.2.1   HYPER-PARAMETERS IN THE PIXEL-BASED SETTING

We use the same hyper-parameters for AARL, SAC (no aug), SAC and CURL during the search phase, to ensure a perfectly fair comparison. When evaluating the searched auxiliary loss, we use a slightly larger setting (e.g., larger batch size) to train RL agents sufficiently. A full list is shown in Table 7.

Table 7: Hyper-parameters used in pixel-based environments.

| Hyperparameter | During Evolution | Final Evaluation of AARL-Pixel |
|---|---|---|
| Random crop | False for SAC (no aug); True for others | True |
| Observation rendering | (84, 84) for SAC (no aug); (100, 100) for others | (100, 100) |
| Observation downsampling | (84, 84) | (84, 84) |
| Replay buffer size | 100000 | 100000 |
| Initial steps | 1000 | 1000 |
| Stacked frames | 3 | 3 |
| Actoin repeat | 4 (Cheetah-Run, Reacher-Easy) 2 (Walker-Walk); | 8 (Cartpole-Swingup); 4 (Others) 2 (Walker-Walk, Finger-Spin) |
| Hidden units (MLP) | 1024 | 1024 |
| Hidden units (Predictor MLP) | 256 | 256 |
| Evaluation episodes | 10 | 10 |
| Optimizer | Adam | Adam |
| $(\beta_1, \beta_2)$ for actor/critic/encoder | (.9, .999) | (.9, .999) |
| $(\beta_1, \beta_2)$ for entropy $\alpha$ | (.5, .999) | (.5, .999) |
| Learning rate for actor/critic | 1e-3 | 2e-4 (Cheetah-Run); 1e-3 (Others) |
| Learning rate for encoder | 1e-3 | 3e-3 (Cheetah-Run, Finger-Spin, Walker-Walk); 1e-3 (Others) |
| Learning for $\alpha$ | 1e-4 | 1e-4 |
| Batch size for RL loss | 128 | 512 |
| Batch size for auxiliary loss | 128 | 128 (Walker-Walk) 256 (Cheetah-Run, Finger-Spin) 512 (Others); |
| Q function EMA $\tau$ | 0.01 | 0.01 |
| Critic target update freq | 2 | 2 |
| Convolutional layers | 4 | 4 |
| Number of filters | 32 | 32 |
| Non-linearity | ReLU | ReLU |
| Encoder EMA $\tau$ | 0.05 | 0.05 |
| Latent dimension | 50 | 50 |
| Discount $\gamma$ | .99 | .99 |
| Initial temperature | 0.1 | 0.1 |

### A.2.2   HYPER-PARAMETERS IN THE STATE-BASED SETTING

We use the same hyper-parameters for AARL, SAC-Identity, SAC-DenseMLP and CURL-DenseMLP, shown in Table 8. As training in state-based environments is substantially faster than pixel-based environments, there is no need to balance training cost and agent performance. We use this setting for both the search phase and the final evaluation phase.

## A.3   EVOLUTION STRATEGY

**Horizon-changing Mutations**   There are two kinds of mutations that are able to change horizon length. One is to decrease horizon length. Specifically, we remove the last time step, i.e.,

Table 8: Hyper-parameters used in state-based environments.

| | |
|---|---|
| Replay buffer size | 100000 |
| Initial steps | 1000 |
| Action repeat | 4 |
| Hidden units (MLP) | 1024 |
| Hidden units (Predictor MLP) | 256 |
| Evaluation episodes | 10 |
| Optimizer | Adam |
| $(\beta_1, \beta_2)$ for actor/critic/encoder | (.9, .999) |
| $(\beta_1, \beta_2)$ for entropy $\alpha$ | (.5, .999) |
| Learning rate for actor/critic/encoder | 2e-4 (Cheetah-Run); 1e-3 (Others) |
| Learning for $\alpha$ | 1e-4 |
| Batch size | 512 |
| Q function EMA $\tau$ | 0.01 |
| Critic target update freq | 2 |
| DenseMLP Layers | 1 |
| Non-linearity | ReLU |
| Encoder EMA $\tau$ | 0.05 |
| Latent dimension of DenseMLP | 40 |
| Discount $\gamma$ | .99 |
| Initial temperature | 0.1 |

$(a_{t+k}, r_{t+k}, s_{t+k+1})$ if the target horizon length is $k$. The other is to increase horizon length, in which we append three randomly generated bits to the given masks at the end. We do not shorten the horizon when it becomes too small (less than 1), or lengthen the horizon when it is too long (exceeding 10).

**Mutating Source and Target Masks**  When mutating a candidate, the mutation on the source and the target masks are independent to each other except for horizon change mutation where two masks could either both increase horizon or decrease horizon.

**Loss-rejection Protocol**  The loss-rejection protocol makes sure that invalid loss functions do not go into the expensive computation of RL training. Concretely, the following conditions must be satisfied to make a valid loss function: 1) having at least one state embedding in $\hat{m}$ to make sure the gradient of auxiliary loss backward propagates to the state encoder, and 2) target $m$ is not empty. If a loss is rejected, we repeat the mutation to fill up the population.

**Initialization**  At each initialization, we randomly generate 100 auxiliary loss functions (every bit of masks are generated from Bernoulli($p$) where $p = 0.5$.) and generate 25 auxiliary loss functions with prior probability, which makes the auxiliary loss have some features like forward dynamics prediction or reward prediction. The prior probability for generating forward dynamics pattern is: 1) every bit of states from target is generated from Bernoulli($p$) where $p = 0.2$; 2) every bit of actions from source is generated from Bernoulli($p$) where $p = 0.8$; 3) every bit of states from target is generated by flipping the states of source; 4) The other bits are generated from Bernoulli($p$) where $p = 0.5$. The prior probability for generating reward prediction pattern is: 1) every bit of rewards from target is generated from Bernoulli($p$) where $p = 0.8$; 2) Every bit of states and actions from target are 0; 3) The other bits are generated from Bernoulli($p$) where $p = 0.5$.

## A.4 CROSS VALIDATION

After the evolutionary search is done, we choose top-5 candidates for cross-validation across environments. This step is to avoid overfitting to one single environment. For each top-5 loss candidate, we run 3 trials on each of these 3 training environments (45 runs in total). Then we sort these top-5 candidates by cross-validation overall performance (mean AULC score on three environments) and retrieve the top-1 candidate as our final searched loss. We do this in pixel-based environments and state-based environments separately. For pixel-based RL, the top-5 candidates are the top candidates

from stage 5 of Cheetah-Run (Pixel). For state-based RL, the top-5 are the top candidates from stage 4 and 5 of Cheetah-Run (State).

## A.5 BASELINES IMPLEMENTATION

**Pixel-based Setting** CURL (Laskin et al., 2020) is the main baseline to compare with in the pixel-based setting, which is considered to be the state-of-the-art pixel-based RL algorithm. CURL learns state representations with a contrastive auxiliary loss. PlaNet (Hafner et al., 2019b) and Dreamer (Hafner et al., 2019a) are model-based methods that generate synthetic rollouts with a learned world model. SAC+AE (Yarats et al., 2019) uses a reconstruction auxiliary loss of images to boost RL training. SLAC (Lee et al., 2019) leverages forward dynamics to construct a latent space for RL agents. Note that there are two versions of SLAC with different gradient Updates per agent step: SLACv1 (1:1) and SLACv2(3:1). We adopt SLACv1 for comparison since all methods only make one gradient update per agent step. Pixel SAC are just vanilla SAC (Haarnoja et al., 2018) agents with images a as inputs respectively.

**State-based Setting** As for the state-based setting, we compare AARL-State with SAC-Identity, SAC and CURL. SAC-Identity is the vanilla state-based SAC where states are directly fed to actor/critic networks. SAC and CURL use the same architecture of 1-layer densely connected MLP as a state encoder. Note that both AARL and baseline methods use the same hyper-parameter reported in Table 8 without additional hyper-parameter tuning.

## B FURTHER EXPERIMENT RESULTS

### B.1 LEARNING CURVES FOR AARL ON PIXEL-BASED DMCONTROL

We benchmark the performance of AARL to the best-performing pixel-based baseline (CURL). As shown in Figure 9, the sample efficiency of AARL outperforms CURL on 10 out of 12 environments. Note that the learning curves of CURL may not match to the data in Table 3, this is because we use the data reported in CURL paper for tabular while we rerun CURL for learning curves plotting, where we find the performance of our rerunning CURL is sightly below the CURL paper.

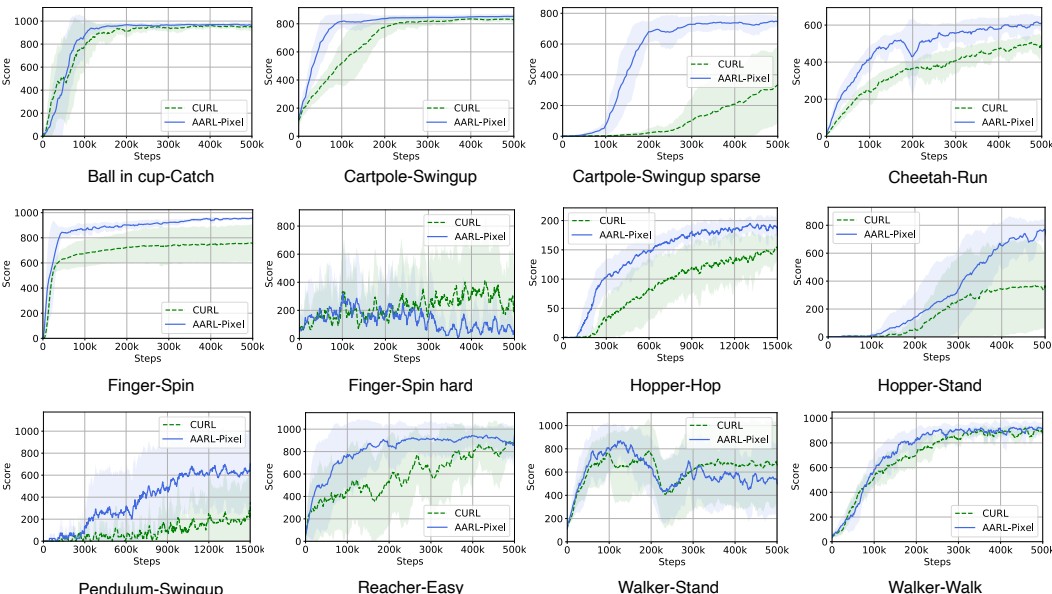

Figure 9: Learning curves of AARL-Pixel and CURL on 12 DMC environments. Shadow represents the standard deviation over five random seeds. The curves are uniformly smoothed for visual display. The y-axis represents episodic reward and x-axis represents interaction steps.

## B.2 Effectiveness of AULC scores

To illustrate intuitively why we use the area under learning curve instead of other metrics, we select top-10 candidates with different evolution metrics. Figure 10 demonstrates the usage of AULC score could well balance both sample efficiency and final performance. The learning curves of the top-10 candidates selected by AULC score look better than the other two metrics (that select top candidates simply with 100k step score or 500k step score).

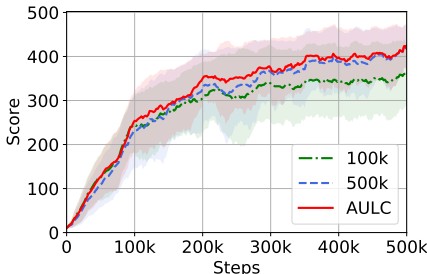

Figure 10: Learning curves of top-10 loss candidates selected with different metrics.

## B.3 Histogram of Auxiliary Loss Analysis

The histogram of each pattern analysis is shown in Figure 11.

## C Search Space Complexity Analysis

The search space size is the size of loss input space multiplied by the size of loss operator space.

For loss input, we calculate separately for each possible horizon length $k$. When length is $k$, the interaction sequence length $(s_t, a_t, r_t, \cdots, s_{t+k})$ has length $(3k + 1)$. For binary mask $\hat{m}$, there are $2^{3k+1}$ different options. There are also $2^{3k+1}$ distinct binary mask $m$ to select targets. Therefore, there are $2^{6k+2}$ combinations when horizon length is fixed to $k$. As our maximum horizon is 10, we enumerate $k$ from 1 to 10, resulting in $\sum_{i=1}^{10} 2^{6i+2}$.

For loss operator, we can learn intuitively from Table 2 that there are 5 different similarity measures with or without negative samples, resulting in $5 \times 2 = 10$ different loss operators.

In total, the size of the entire space is

$$10 \times \sum_{i=1}^{10} 2^{6i+2} \approx 4.6 \times 10^{19}.$$

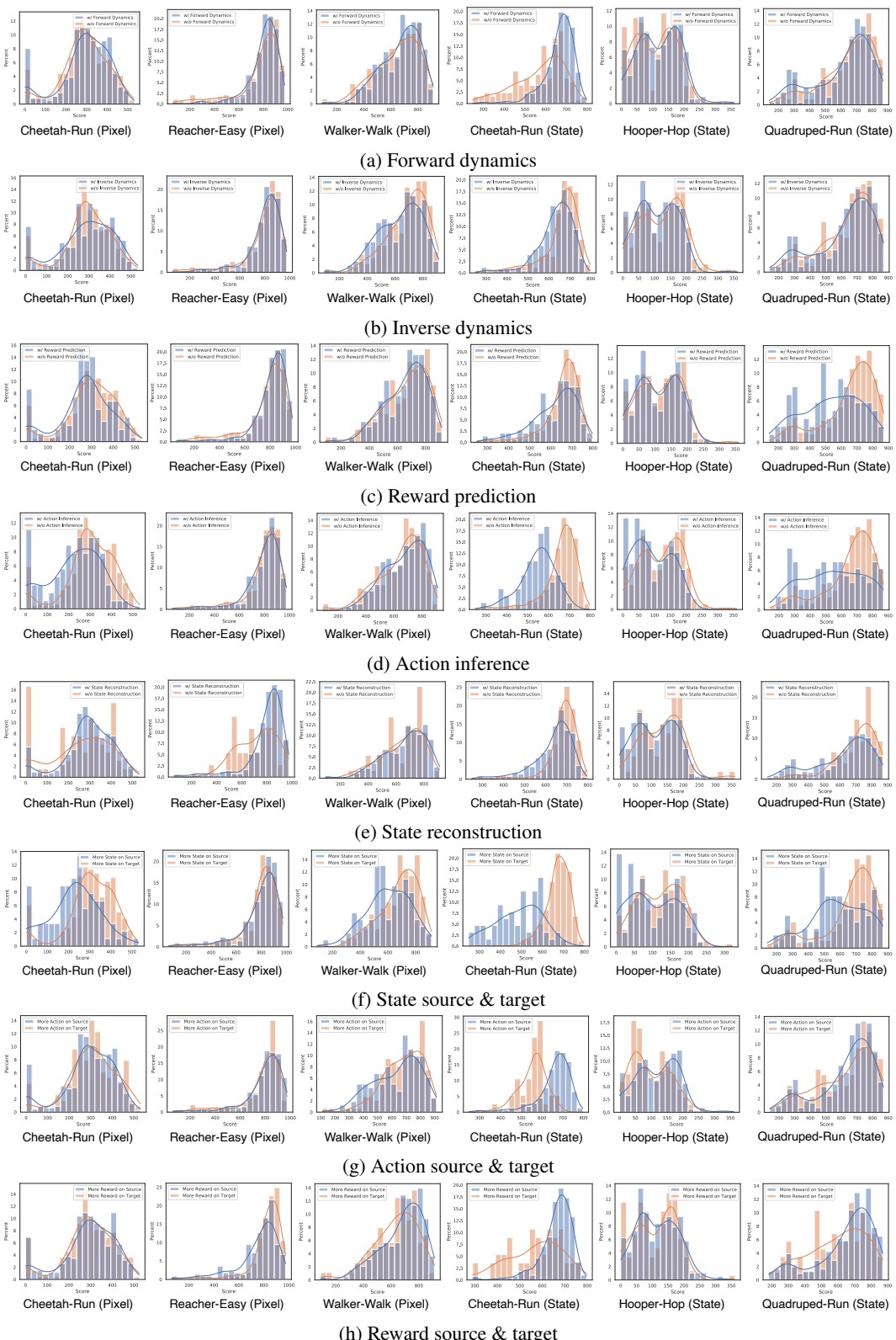

Figure 11: Histogram of statistical analysis of auxiliary loss candidates in 6 evolution process. The x-axis represents approximated AULC score while the y-axis represents the percentage of corresponding bin of population. Best viewed in color.

# D  TOP-PERFORMING AUXILIARY LOSSES

## D.1  AARL-PIXEL AND AARL-STATE

The auxiliary loss of AARL-Pixel is:

$$\{s_{t+1}, a_{t+1}, a_{t+2}, a_{t+3}\} \rightarrow \{r_t, r_{t+1}, s_{t+2}, s_{t+3}\} \tag{4}$$

which is the third-best candidate of stage 4 in Cheetah-Run (Pixel).

The auxiliary loss of AARL-State is:

$$\begin{aligned}\{s_t, a_t, a_{t+1}, s_{t+2}, a_{t+2}, a_{t+3}, r_{t+3}, a_{t+4}, r_{t+4}, a_{t+5}, a_{t+7}, s_{t+8}, a_{t+8}, r_{t+8}\} \\ \rightarrow \{s_{t+1}, s_{t+3}, a_{t+4}, s_{t+6}, s_{t+9}\}\end{aligned} \tag{5}$$

which is the fourth-best candidate of stage 4 in Cheetah-Run (State).

These two losses are chosen because they are the best-performing loss functions during cross-validation (Appendix A.4).

## D.2  DURING EVOLUTION

We report all the top-5 auxiliary loss candidates during evolution in this section.

Table 9: Top-5 candidates of each stage in Cheetah-Run (Pixel) evolution process

| | Cheetah-Run (Pixel) |
|---|---|
| Stage-1 | $\{r_t, s_{t+1}, a_{t+1}, r_{t+1}, a_{t+2}, r_{t+2}, a_{t+3}, r_{t+3}\} \rightarrow \{s_t, a_t, s_{t+2}, s_{t+3}, s_{t+4}\}$ 
 $\{s_t, a_t, r_t\} \rightarrow \{s_{t+1}\}$ 
 $\{s_t, a_t, a_{t+1}, r_{t+2}\} \rightarrow \{s_t, a_t, s_{t+1}, a_{t+1}, r_{t+1}, s_{t+2}, r_{t+2}, s_{t+3}\}$ 
 $\{s_t, r_t, a_{t+1}, a_{t+2}, a_{t+3}, r_{t+3}, r_{t+4}, a_{t+5}, r_{t+5}, s_{t+6}, s_{t+7}\} \rightarrow \{s_t, a_t, s_{t+1}, s_{t+2}, r_{t+2}, r_{t+3}, s_{t+4}, r_{t+5}, s_{t+6}, a_{t+6}, s_{t+7}\}$ 
 $\{s_t, a_t, s_{t+1}, a_{t+1}, s_{t+2}, r_{t+2}\} \rightarrow \{s_t, s_{t+1}, r_{t+1}, s_{t+2}, r_{t+2}, s_{t+3}\}$ |
| Stage-2 | $\{s_t, a_{t+1}, r_{t+2}, s_{t+4}, r_{t+4}\} \rightarrow \{s_{t+2}, a_{t+3}, r_{t+3}, a_{t+4}, s_{t+5}\}$ 
 $\{s_t, a_t, a_{t+2}, r_{t+2}\} \rightarrow \{s_t, r_t, s_{t+1}, s_{t+2}, r_{t+2}\}$ 
 $\{a_t, r_t, s_{t+1}, r_{t+1}, s_{t+2}, a_{t+2}, r_{t+2}, a_{t+3}, a_{t+4}\} \rightarrow \{s_{t+1}, s_{t+2}, s_{t+3}, a_{t+3}, s_{t+4}\}$ 
 $\{s_t, a_t, r_t, a_{t+1}, r_{t+1}, a_{t+2}, r_{t+2}, a_{t+3}, a_{t+4}\} \rightarrow \{s_t, s_{t+1}, s_{t+2}, s_{t+4}, s_{t+5}\}$ 
 $\{r_t, s_{t+1}, r_{t+1}\} \rightarrow \{s_t, a_t, a_{t+1}, s_{t+2}\}$ |
| Stage-3 | $\{s_t, a_t, a_{t+2}, r_{t+2}\} \rightarrow \{s_t, s_{t+1}, s_{t+2}, r_{t+2}\}$ 
 $\{s_t, r_t, a_{t+1}, a_{t+3}, r_{t+3}, r_{t+4}, a_{t+5}, r_{t+5}, s_{t+6}, a_{t+6}, s_{t+7}\} \rightarrow \{s_t, a_t, s_{t+1}, s_{t+2}, r_{t+2}, r_{t+3}, s_{t+4}, r_{t+5}, s_{t+6}, s_{t+7}, a_{t+7}\}$ 
 $\{s_t, a_t, a_{t+1}, r_{t+1}, a_{t+2}, r_{t+2}, a_{t+3}, s_{t+4}, a_{t+4}\} \rightarrow \{s_{t+1}, s_{t+2}, s_{t+4}, s_{t+5}\}$ 
 $\{s_t, a_t, a_{t+1}, r_{t+1}, r_{t+2}, s_{t+3}, a_{t+3}, r_{t+4}\} \rightarrow \{s_{t+1}, s_{t+2}, r_{t+3}, s_{t+4}, a_{t+4}, s_{t+5}\}$ 
 $\{r_t, s_{t+1}\} \rightarrow \{s_t, a_t, a_{t+1}, s_{t+2}\}$ |
| Stage-4 | $\{s_t, s_{t+1}, a_{t+2}, r_{t+2}, s_{t+3}, s_{t+4}\} \rightarrow \{a_{t+1}, s_{t+2}, r_{t+2}, s_{t+4}, a_{t+4}, s_{t+5}\}$ 
 $\{s_t, a_t, a_{t+1}, r_{t+1}, r_{t+2}, s_{t+3}, a_{t+3}\} \rightarrow \{s_{t+1}, s_{t+2}, r_{t+3}, s_{t+4}\}$ 
 $\{s_t\} \rightarrow \{s_t, r_t, s_{t+1}, r_{t+1}, s_{t+2}, a_{t+2}, r_{t+2}\}$ 
 $\{s_t, r_t, a_{t+1}, s_{t+2}, a_{t+2}, r_{t+2}, a_{t+3}, r_{t+3}, a_{t+4}\} \rightarrow \{s_t, a_t, s_{t+1}, r_{t+1}, r_{t+3}, s_{t+4}, a_{t+4}, s_{t+5}\}$ 
 $\{r_t, s_{t+1}, a_{t+1}\} \rightarrow \{s_t, a_t, a_{t+1}, s_{t+2}\}$ |
| Stage-5* | $\{a_t, r_t, s_{t+1}, a_{t+1}, r_{t+1}, a_{t+2}, r_{t+2}, a_{t+3}, r_{t+3}\} \rightarrow \{r_t, s_{t+1}, a_{t+1}, s_{t+2}, s_{t+4}\}$ 
 $\{s_t, a_{t+1}, r_{t+2}, a_{t+3}, s_{t+4}, r_{t+4}\} \rightarrow \{s_{t+1}, s_{t+2}, a_{t+3}, s_{t+4}, a_{t+4}, s_{t+5}\}$ 
 $^{\dagger}\{s_{t+1}, a_{t+1}, a_{t+2}, a_{t+3}\} \rightarrow \{r_t, r_{t+1}, s_{t+2}, s_{t+3}\}$ 
 $\{s_t\} \rightarrow \{s_t, r_t, s_{t+1}, r_{t+1}, s_{t+2}\}$ 
 $\{s_t\} \rightarrow \{s_t, r_t, s_{t+1}, r_{t+1}, s_{t+2}, r_{t+2}\}$ |
| Stage-6 | $\{a_{t+1}, r_{t+1}, s_{t+2}, r_{t+2}, a_{t+3}, a_{t+4}\} \rightarrow \{r_t, s_{t+1}, r_{t+1}, r_{t+3}, s_{t+4}, a_{t+4}, s_{t+5}\}$ 
 $\{s_t, a_{t+1}, a_{t+3}, s_{t+4}, r_{t+4}\} \rightarrow \{s_{t+1}, s_{t+2}, a_{t+3}, s_{t+4}, a_{t+4}, s_{t+5}\}$ 
 $\{s_t, a_{t+1}, r_{t+1}, s_{t+2}, a_{t+2}, r_{t+2}, s_{t+3}, a_{t+3}, r_{t+3}\} \rightarrow \{a_t, s_{t+2}, r_{t+2}, a_{t+3}, s_{t+4}, a_{t+4}, s_{t+5}\}$ 
 $\{s_t, a_{t+1}, r_{t+2}, a_{t+3}, s_{t+4}, r_{t+4}\} \rightarrow \{s_{t+1}, s_{t+2}, a_{t+3}, s_{t+4}, a_{t+4}, s_{t+5}, a_{t+5}\}$ 
 $\{s_t, a_{t+1}, a_{t+2}, r_{t+2}, s_{t+3}, a_{t+3}, s_{t+4}, r_{t+4}\} \rightarrow \{s_{t+1}, a_{t+1}, s_{t+2}, r_{t+2}, s_{t+4}, a_{t+4}, r_{t+4}, s_{t+5}\}$ |
| Stage-7 | $\{s_t, a_t, r_t, a_{t+1}, r_{t+2}, a_{t+3}, s_{t+4}\} \rightarrow \{a_t, r_t, s_{t+2}, r_{t+3}, s_{t+4}, a_{t+4}, r_{t+4}, s_{t+5}\}$ 
 $\{s_t, r_{t+2}, a_{t+3}, s_{t+4}\} \rightarrow \{a_t, s_{t+1}, s_{t+2}, r_{t+2}, a_{t+3}, r_{t+3}, s_{t+4}, a_{t+4}, r_{t+4}, s_{t+5}\}$ 
 $\{s_{t+1}, a_{t+2}, r_{t+2}, s_{t+3}, a_{t+3}, r_{t+3}, s_{t+4}\} \rightarrow \{r_t, a_{t+1}, r_{t+1}, s_{t+2}, r_{t+2}, s_{t+4}, a_{t+4}, s_{t+5}\}$ 
 $\{s_t, a_{t+1}, s_{t+2}, a_{t+2}, r_{t+2}, s_{t+3}, a_{t+3}, r_{t+3}, a_{t+4}\} \rightarrow \{a_t, s_{t+1}, s_{t+2}, r_{t+2}, s_{t+3}, r_{t+3}, s_{t+4}, a_{t+4}, s_{t+5}\}$ 
 $\{s_t, a_{t+1}, r_{t+2}, a_{t+3}, s_{t+4}, r_{t+4}\} \rightarrow \{s_{t+1}, s_{t+2}, r_{t+2}, a_{t+3}, s_{t+4}, a_{t+4}, s_{t+5}, a_{t+5}\}$ |

∗: Used for cross-validation. †: AARL-Pixel.

Table 10: Top-5 candidates of each stage in Reacher-Easy (Pixel) evolution process

| | Reacher-Easy (Pixel) |
|---|---|
| Stage-1 | $\{s_{t+1}, a_{t+1}\} \to \{r_t, r_{t+1}\}$
$\{r_t, s_{t+1}, a_{t+1}, s_{t+2}, a_{t+2}, r_{t+2}, a_{t+3}, a_{t+4}, r_{t+4}, s_{t+5}, a_{t+5}, r_{t+5}, s_{t+6}, a_{t+6}, r_{t+6}, r_{t+7}, s_{t+8}, a_{t+8}, s_{t+9}\} \to \{a_{t+1}, r_{t+1}, s_{t+2}, s_{t+3}, r_{t+3}, s_{t+4}, a_{t+4}, r_{t+4}, a_{t+5}, a_{t+6}, r_{t+6}, s_{t+7}, a_{t+7}, r_{t+7}, a_{t+8}, r_{t+8}, s_{t+10}\}$
$\{s_t, r_t, s_{t+1}, a_{t+2}, s_{t+4}, a_{t+4}, r_{t+5}, s_{t+6}, a_{t+6}, a_{t+7}, s_{t+9}, s_{t+10}\} \to \{s_t, s_{t+1}, r_{t+1}, s_{t+2}, r_{t+2}, r_{t+3}, s_{t+4}, a_{t+4}, r_{t+4}, a_{t+5}, s_{t+6}, a_{t+6}, r_{t+7}, r_{t+8}, s_{t+10}\}$
$\{s_t, r_t, s_{t+1}, a_{t+1}, s_{t+2}, s_{t+3}, a_{t+3}, a_{t+4}, s_{t+5}, r_{t+5}, a_{t+6}, r_{t+6}, s_{t+7}\} \to \{a_{t+1}, s_{t+2}, a_{t+2}, r_{t+2}, s_{t+3}, a_{t+3}, r_{t+3}, s_{t+4}, a_{t+4}, r_{t+4}, s_{t+6}, a_{t+6}\}$
$\{s_{t+1}, r_{t+1}, a_{t+2}, r_{t+2}, r_{t+3}, s_{t+4}, a_{t+4}, r_{t+4}, s_{t+5}, s_{t+6}, r_{t+6}\} \to \{r_t, s_{t+1}, s_{t+2}, a_{t+2}, r_{t+2}, s_{t+4}, s_{t+5}, r_{t+5}, s_{t+6}, a_{t+6}, s_{t+7}, s_{t+8}\}$ |
| Stage-2 | $\{s_t, s_{t+1}, a_{t+1}\} \to \{r_t, r_{t+1}\}$
$\{s_t, r_t, a_{t+2}, a_{t+3}, s_{t+4}, a_{t+4}, r_{t+4}, a_{t+5}, r_{t+5}, s_{t+6}, a_{t+7}, a_{t+8}\} \to \{r_t, a_{t+1}, s_{t+2}, r_{t+2}, s_{t+3}, a_{t+3}, r_{t+3}, a_{t+4}, r_{t+4}, a_{t+5}, r_{t+5}, a_{t+6}, s_{t+7}, s_{t+9}\}$
$\{s_t, a_t, r_t, s_{t+1}, a_{t+2}, a_{t+3}, s_{t+5}\} \to \{a_t, s_{t+2}, a_{t+2}, r_{t+2}, s_{t+3}, a_{t+3}, a_{t+4}, r_{t+4}\}$
$\{s_t, a_t, s_{t+1}, a_{t+3}, s_{t+4}, a_{t+4}, a_{t+5}, s_{t+6}, a_{t+6}, a_{t+7}, r_{t+7}, s_{t+9}, a_{t+9}\} \to \{s_t, a_t, r_t, s_{t+1}, r_{t+1}, s_{t+2}, r_{t+3}, s_{t+4}, r_{t+4}, s_{t+6}, a_{t+6}, r_{t+7}, a_{t+8}, s_{t+10}\}$
$\{s_t, a_t, r_t, a_{t+1}, a_{t+3}, r_{t+3}, s_{t+4}, a_{t+4}, s_{t+5}, a_{t+5}, r_{t+5}, a_{t+6}, r_{t+6}, a_{t+7}\} \to \{s_{t+1}, s_{t+2}, a_{t+2}, s_{t+3}, s_{t+6}, a_{t+6}, s_{t+7}, s_{t+8}\}$ |
| Stage-3 | $\{a_t, s_{t+1}, a_{t+1}, s_{t+2}, a_{t+2}\} \to \{r_t, a_{t+1}, r_{t+1}, r_{t+2}\}$
$\{s_t, a_t, s_{t+1}, s_{t+2}, a_{t+2}, a_{t+3}, s_{t+4}, r_{t+4}, a_{t+5}, r_{t+5}, r_{t+6}, a_{t+7}\} \to \{s_t, a_t, r_t, s_{t+2}, s_{t+3}, a_{t+3}, s_{t+4}, a_{t+4}, a_{t+5}, r_{t+5}, s_{t+6}, s_{t+7}, s_{t+8}\}$
$\{s_t, a_t, s_{t+1}, a_{t+1}, r_{t+1}, s_{t+2}, a_{t+2}, a_{t+3}, s_{t+4}, r_{t+4}, s_{t+6}, a_{t+7}\} \to \{r_t, a_{t+2}, r_{t+2}, a_{t+3}, r_{t+3}, s_{t+4}, a_{t+4}, a_{t+5}, s_{t+6}, a_{t+6}, s_{t+7}, r_{t+7}, s_{t+8}\}$
$\{s_t, a_t, s_{t+1}, r_{t+1}, a_{t+2}, a_{t+3}, s_{t+4}, r_{t+4}, s_{t+6}, r_{t+6}, a_{t+7}\} \to \{s_t, r_t, r_{t+1}, a_{t+2}, r_{t+2}, a_{t+3}, r_{t+3}, s_{t+4}, a_{t+4}, a_{t+5}, r_{t+5}, s_{t+6}, s_{t+7}, s_{t+8}\}$
$\{s_t, a_t, r_t, a_{t+1}, r_{t+1}, a_{t+2}, s_{t+6}, s_{t+7}, a_{t+7}, s_{t+8}\} \to \{s_t, r_t, s_{t+2}, a_{t+2}, r_{t+2}, a_{t+3}, r_{t+3}, s_{t+8}, a_{t+8}\}$ |
| Stage-4 | $\{s_t, a_t, r_t, a_{t+1}, a_{t+3}, r_{t+3}, s_{t+4}, s_{t+5}, a_{t+5}, r_{t+5}, a_{t+6}, r_{t+6}, a_{t+7}\} \to \{s_{t+1}, s_{t+2}, s_{t+3}, r_{t+4}, r_{t+5}, r_{t+7}, s_{t+8}\}$
$\{a_t, s_{t+1}, a_{t+1}, s_{t+2}\} \to \{r_t, r_{t+1}, a_{t+2}, r_{t+2}\}$
$\{s_t, a_t, r_t, a_{t+1}, r_{t+3}, s_{t+4}, s_{t+5}, s_{t+6}, r_{t+6}, a_{t+7}\} \to \{s_{t+1}, s_{t+2}, a_{t+3}, s_{t+4}, r_{t+4}, s_{t+5}, s_{t+6}, s_{t+7}, r_{t+7}, s_{t+8}\}$
$\{s_t, a_t, r_{t+1}, a_{t+2}, a_{t+3}, s_{t+4}, r_{t+4}, r_{t+5}, s_{t+6}, r_{t+6}, a_{t+7}\} \to \{s_t, r_t, r_{t+1}, a_{t+2}, r_{t+2}, a_{t+3}, r_{t+3}, s_{t+4}, a_{t+4}, a_{t+5}, r_{t+5}, s_{t+6}, s_{t+7}, s_{t+8}\}$
$\{a_t, s_{t+1}, a_{t+1}, s_{t+2}, a_{t+2}\} \to \{r_t, r_{t+1}, r_{t+2}\}$ |
| Stage-5 | $\{s_t, a_t, a_{t+1}, a_{t+2}, a_{t+3}, s_{t+4}, s_{t+5}, a_{t+6}, r_{t+6}, a_{t+7}, s_{t+8}\} \to \{r_t, a_{t+2}, r_{t+2}, a_{t+3}, s_{t+4}, s_{t+5}, a_{t+5}, s_{t+6}, a_{t+6}, s_{t+7}, r_{t+7}, s_{t+8}\}$
$\{s_t, a_t, s_{t+1}, a_{t+1}, r_{t+1}\} \to \{r_t, s_{t+2}\}$
$\{s_t, a_t, r_t, a_{t+1}, r_{t+3}, s_{t+4}, s_{t+5}, s_{t+6}, r_{t+6}, a_{t+7}\} \to \{s_{t+1}, s_{t+2}, a_{t+3}, s_{t+4}, r_{t+4}, s_{t+5}, r_{t+6}, s_{t+7}, r_{t+7}, s_{t+8}\}$
$\{s_t, a_t, r_t, a_{t+1}, r_{t+1}, s_{t+2}, a_{t+2}, a_{t+3}, s_{t+4}, s_{t+5}, a_{t+5}, r_{t+6}, a_{t+7}\} \to \{s_t, r_t, a_{t+1}, r_{t+1}, a_{t+2}, r_{t+2}, s_{t+3}, a_{t+3}, s_{t+4}, a_{t+4}, s_{t+6}, a_{t+6}, r_{t+6}, s_{t+7}\}$
$\{s_t, a_t, s_{t+1}, r_{t+1}, a_{t+2}, s_{t+3}, s_{t+4}, r_{t+6}, a_{t+7}, s_{t+8}\} \to \{s_t, r_t, r_{t+1}, a_{t+2}, r_{t+2}, a_{t+3}, s_{t+4}, a_{t+4}, a_{t+5}, s_{t+6}, r_{t+6}, s_{t+7}, s_{t+8}\}$ |
| Stage-6 | $\{s_t, a_t, r_t, a_{t+1}, a_{t+3}, r_{t+3}, s_{t+4}, a_{t+4}, s_{t+5}, a_{t+5}, r_{t+5}, s_{t+6}, a_{t+6}, r_{t+6}\} \to \{s_{t+1}, s_{t+2}, r_{t+2}, s_{t+3}, a_{t+4}, s_{t+6}\}$
$\{s_t, a_t, r_t, s_{t+1}, a_{t+1}, r_{t+1}, s_{t+2}, a_{t+2}, s_{t+3}, a_{t+4}, r_{t+5}, r_{t+6}, s_{t+7}, s_{t+8}\} \to \{s_t, r_t, r_{t+2}, a_{t+3}, r_{t+3}, s_{t+4}, r_{t+4}, a_{t+5}, r_{t+5}, s_{t+6}, a_{t+6}, r_{t+6}, s_{t+7}, a_{t+7}, r_{t+7}, s_{t+8}\}$
$\{s_t, a_t, r_t, r_{t+1}, a_{t+2}, a_{t+3}, s_{t+4}, s_{t+5}, r_{t+5}\} \to \{s_t, r_t, r_{t+1}, a_{t+2}, a_{t+3}, r_{t+3}, a_{t+4}, r_{t+4}, a_{t+5}, s_{t+6}, a_{t+6}, r_{t+6}, s_{t+7}, s_{t+8}\}$
$\{s_t, a_t, s_{t+1}, a_{t+3}, a_{t+4}, a_{t+5}, s_{t+6}, a_{t+6}, s_{t+7}, a_{t+7}, r_{t+7}, s_{t+9}, a_{t+9}\} \to \{s_t, a_t, r_t, s_{t+1}, r_{t+1}, s_{t+2}, s_{t+3}, r_{t+3}, s_{t+4}, a_{t+4}, r_{t+4}, s_{t+6}, a_{t+6}, r_{t+7}, a_{t+8}, a_{t+9}, s_{t+10}\}$
$\{s_t, a_t, r_t, a_{t+1}, r_{t+1}, a_{t+2}, a_{t+3}, s_{t+4}, s_{t+5}, a_{t+5}, r_{t+6}, a_{t+6}, a_{t+7}\} \to \{r_t, s_{t+1}, r_{t+1}, a_{t+2}, r_{t+2}, s_{t+3}, s_{t+4}, s_{t+6}, a_{t+6}, r_{t+6}, s_{t+7}, s_{t+8}\}$ |
| Stage-7 | $\{s_t, a_t, s_{t+1}, a_{t+1}, r_{t+1}, s_{t+2}, a_{t+2}, r_{t+5}, r_{t+6}, a_{t+7}, s_{t+8}\} \to \{r_t, a_{t+2}, r_{t+2}, r_{t+3}, s_{t+4}, a_{t+4}, a_{t+5}, a_{t+6}, r_{t+6}, s_{t+7}, s_{t+8}\}$
$\{s_t, a_t, a_{t+1}, s_{t+2}, a_{t+2}, a_{t+3}, s_{t+4}, a_{t+5}, r_{t+5}, r_{t+6}, a_{t+6}, a_{t+7}\} \to \{s_{t+1}, r_{t+2}, a_{t+3}, r_{t+3}, s_{t+4}, a_{t+4}, a_{t+5}, s_{t+6}, a_{t+6}, s_{t+7}, r_{t+7}\}$
$\{s_t, a_t, r_t, a_{t+2}, a_{t+3}, s_{t+4}, s_{t+5}, r_{t+5}, a_{t+6}\} \to \{s_t, r_t, r_{t+1}, s_{t+2}, a_{t+2}, a_{t+3}, r_{t+3}, r_{t+4}, s_{t+6}, s_{t+8}\}$
$\{s_t, a_t, r_t, a_{t+1}, a_{t+3}, r_{t+3}, s_{t+4}, a_{t+4}, s_{t+5}, a_{t+5}, r_{t+5}, s_{t+6}, a_{t+6}, r_{t+6}\} \to \{r_t, s_{t+1}, s_{t+2}, a_{t+2}, s_{t+3}, r_{t+3}, s_{t+4}, s_{t+6}\}$
$\{s_t, a_t, r_t, a_{t+1}, r_{t+1}, a_{t+3}, r_{t+3}, s_{t+4}, a_{t+4}, s_{t+5}, r_{t+5}, s_{t+6}, a_{t+7}, r_{t+7}\} \to \{s_t, r_t, s_{t+1}, r_{t+1}, a_{t+2}, s_{t+3}, a_{t+3}, r_{t+4}, s_{t+6}, s_{t+8}\}$ |

Table 11: Top-5 candidates of each stage in Walker-Walk (Pixel) evolution process

| | Walker-Walk (Pixel) |
|---|---|
| Stage-1 | $\{s_t, a_t, s_{t+2}, a_{t+2}, r_{t+2}, s_{t+3}, a_{t+4}, a_{t+5}, s_{t+6}, a_{t+6}, a_{t+7}, s_{t+8}, r_{t+8}\} \to \{s_t, s_{t+1}, r_{t+1}, s_{t+2}, r_{t+2}, r_{t+3}, a_{t+4}, r_{t+4}, a_{t+5}, r_{t+5}, a_{t+6}, r_{t+6}, r_{t+7}, s_{t+8}, a_{t+8}, s_{t+9}\}$
$\{s_t, a_t, a_{t+1}, r_{t+1}\} \to \{a_t, s_{t+1}, r_{t+1}\}$
$\{r_t, a_{t+1}, s_{t+2}, a_{t+2}, r_{t+3}, s_{t+5}, a_{t+5}, a_{t+6}, r_{t+7}, a_{t+8}\} \to \{s_t, a_t, s_{t+1}, s_{t+3}, a_{t+3}, s_{t+4}, a_{t+4}, s_{t+6}, s_{t+7}, a_{t+7}, s_{t+8}, s_{t+9}, a_{t+9}, s_{t+10}\}$
$\{s_t, r_t, s_{t+1}, a_{t+1}, s_{t+2}, a_{t+2}, r_{t+2}, s_{t+3}, a_{t+4}, r_{t+4}, s_{t+5}, a_{t+5}, a_{t+6}\} \to \{s_t, a_t, r_t, s_{t+1}, s_{t+2}, s_{t+3}, s_{t+4}, r_{t+4}, a_{t+5}, r_{t+5}, s_{t+6}, s_{t+7}\}$
$\{s_t, r_t, a_{t+1}, s_{t+2}, a_{t+2}, s_{t+3}, a_{t+3}, a_{t+4}, s_{t+5}, a_{t+5}, r_{t+5}\} \to \{s_t, a_t, r_t, a_{t+1}, r_{t+1}, s_{t+2}, s_{t+3}, a_{t+3}, s_{t+5}, a_{t+5}, r_{t+5}, s_{t+6}\}$ |
| Stage-2 | $\{s_t, r_t, s_{t+1}, a_{t+1}, r_{t+1}, s_{t+3}, a_{t+3}, a_{t+4}\} \to \{a_t, r_t, s_{t+1}, s_{t+2}, s_{t+3}, r_{t+3}, a_{t+4}, s_{t+5}\}$
$\{s_t, r_t, s_{t+2}, a_{t+2}, r_{t+2}, s_{t+3}, r_{t+3}\} \to \{a_t, s_{t+1}, a_{t+1}, s_{t+2}, a_{t+2}, r_{t+2}, r_{t+3}, s_{t+4}\}$
$\{r_t, a_{t+1}, s_{t+2}, r_{t+2}, s_{t+3}, r_{t+3}\} \to \{a_t, s_{t+1}, a_{t+1}, s_{t+2}, a_{t+2}, r_{t+2}, a_{t+3}, r_{t+3}, s_{t+4}\}$
$\{s_t, r_t, s_{t+2}, s_{t+3}, r_{t+3}, s_{t+4}\} \to \{s_t, r_t, s_{t+1}, a_{t+2}, r_{t+2}, s_{t+3}, s_{t+4}, a_{t+4}, s_{t+5}\}$
$\{s_t, r_t, s_{t+1}, s_{t+2}, s_{t+4}, a_{t+4}, a_{t+5}, s_{t+6}, r_{t+6}, s_{t+7}\} \to \{s_t, r_t, a_{t+1}, r_{t+1}, s_{t+2}, r_{t+2}, s_{t+3}, s_{t+4}, a_{t+4}, r_{t+5}, s_{t+6}, a_{t+7}, r_{t+7}, s_{t+8}\}$ |
| Stage-3 | $\{s_t, a_t, s_{t+1}, a_{t+1}, r_{t+1}, s_{t+2}, r_{t+2}, s_{t+3}, a_{t+3}, r_{t+3}\} \to \{s_{t+2}, a_{t+2}, s_{t+3}, r_{t+3}, s_{t+4}\}$
$\{s_t, r_t, s_{t+1}, a_{t+1}, r_{t+1}, s_{t+2}, r_{t+2}, a_{t+3}, s_{t+4}, a_{t+4}\} \to \{a_t, r_t, s_{t+1}, s_{t+2}, a_{t+2}, s_{t+3}, r_{t+3}, a_{t+4}, s_{t+5}\}$
$\{s_t, a_t, a_{t+1}, r_{t+1}\} \to \{s_{t+2}\}$
$\{s_t, r_t, s_{t+1}, a_{t+1}, r_{t+1}, s_{t+3}, a_{t+3}, a_{t+4}, r_{t+4}\} \to \{s_t, a_t, r_t, s_{t+1}, s_{t+2}, a_{t+2}, s_{t+3}, r_{t+3}, a_{t+4}, s_{t+5}\}$
$\{s_t, r_t, s_{t+2}, a_{t+2}, r_{t+2}, s_{t+3}, r_{t+3}\} \to \{a_t, s_{t+1}, a_{t+1}, s_{t+2}, a_{t+2}, r_{t+2}, r_{t+3}, s_{t+4}\}$ |
| Stage-4 | $\{s_t, a_t, a_{t+1}\} \to \{s_{t+1}, a_{t+1}, s_{t+2}\}$
$\{s_t, r_t, r_{t+1}, s_{t+2}, s_{t+3}, r_{t+3}, r_{t+4}\} \to \{s_t, a_t, r_t, s_{t+1}, r_{t+1}, s_{t+2}, a_{t+2}, s_{t+3}, r_{t+3}, s_{t+4}, a_{t+4}, s_{t+5}\}$
$\{s_t, s_{t+2}, s_{t+3}, a_{t+3}, r_{t+3}, s_{t+4}, a_{t+4}\} \to \{s_t, a_t, r_t, a_{t+2}, r_{t+2}, a_{t+4}, s_{t+5}\}$
$\{s_t, r_t, s_{t+1}, a_{t+1}, r_{t+1}, s_{t+2}, a_{t+3}, s_{t+4}, a_{t+4}\} \to \{a_t, r_t, s_{t+1}, s_{t+2}, a_{t+2}, r_{t+3}, s_{t+5}\}$
$\{s_t, r_t, s_{t+1}, a_{t+1}, r_{t+1}, r_{t+2}, s_{t+3}, a_{t+3}, s_{t+4}, a_{t+4}\} \to \{a_t, r_t, s_{t+1}, s_{t+2}, a_{t+2}, s_{t+3}, r_{t+3}, a_{t+4}, s_{t+5}\}$ |

Table 12: Top-5 candidates of each stage in Cheetah-Run (State) evolution process

| | Cheetah-Run (Raw) |
|---|---|
| Stage-1 | $\{s_t, a_t, r_t, a_{t+1}, r_{t+1}\} \rightarrow \{s_{t+1}, s_{t+2}\}$ 
 $\{a_t, r_t, s_{t+2}, a_{t+2}, a_{t+3}, r_{t+3}\} \rightarrow \{s_t, s_{t+1}, a_{t+1}, s_{t+3}, s_{t+4}\}$ 
 $\{a_t, a_{t+1}, s_{t+2}, a_{t+2}, r_{t+2}, a_{t+3}, r_{t+3}, s_{t+4}, r_{t+4}, a_{t+5}, r_{t+5}, a_{t+7}, r_{t+7}, s_{t+8}, a_{t+8}, r_{t+8}\} \rightarrow \{s_t, s_{t+1}, s_{t+3}, a_{t+4}, s_{t+5}, s_{t+6}, a_{t+6}, s_{t+7}, s_{t+9}\}$ 
 $\{a_{t+1}, a_{t+2}, s_{t+3}, a_{t+3}, a_{t+4}, a_{t+5}, r_{t+5}, a_{t+6}, r_{t+7}\} \rightarrow \{s_t, a_t, s_{t+1}, s_{t+2}, s_{t+4}, s_{t+5}, s_{t+6}, s_{t+7}, a_{t+7}, s_{t+8}\}$ 
 $\{s_t, a_t, a_{t+1}, a_{t+2}, r_{t+3}, a_{t+4}, r_{t+4}, s_{t+5}, a_{t+5}, a_{t+6}, s_{t+7}, a_{t+7}, s_{t+8}, a_{t+8}, r_{t+8}\} \rightarrow \{s_{t+1}, s_{t+2}, s_{t+3}, a_{t+3}, s_{t+4}, s_{t+6}, s_{t+9}\}$ |
| Stage-2 | $\{s_t, a_t, r_t, a_{t+1}, r_{t+1}\} \rightarrow \{s_{t+1}, s_{t+2}\}$ 
 $\{s_t, a_t, r_t, a_{t+1}, r_{t+1}\} \rightarrow \{s_{t+1}, s_{t+2}\}$ 
 $\{s_t, a_t, a_{t+1}, r_{t+1}, a_{t+2}, r_{t+2}, a_{t+3}, r_{t+3}, a_{t+4}, r_{t+4}, s_{t+5}, a_{t+5}, r_{t+5}, a_{t+6}, a_{t+7}, a_{t+8}, r_{t+8}\} \rightarrow \{a_t, s_{t+1}, s_{t+2}, a_{t+2}, s_{t+3}, s_{t+4}, s_{t+6}, s_{t+9}\}$ 
 $\{s_t, a_t, a_{t+1}, s_{t+2}, a_{t+2}, a_{t+3}, r_{t+3}, a_{t+4}, r_{t+4}, a_{t+5}, a_{t+7}, s_{t+8}, a_{t+8}, r_{t+8}\} \rightarrow \{s_{t+1}, s_{t+3}, a_{t+4}, s_{t+6}, s_{t+9}\}$ 
 $\{s_t, a_t, r_t\} \rightarrow \{r_t, s_{t+1}\}$ |
| Stage-3 | $\{s_t, a_t, r_t, a_{t+1}, r_{t+1}\} \rightarrow \{s_{t+1}\}$ 
 $\{s_t, a_t\} \rightarrow \{r_t, s_{t+1}\}$ 
 $\{s_t, a_t, r_t, a_{t+1}\} \rightarrow \{s_{t+2}\}$ 
 $\{s_t, a_t, r_t\} \rightarrow \{s_{t+1}\}$ 
 $\{s_t, a_t, a_{t+1}, r_{t+1}\} \rightarrow \{s_t, s_{t+1}, s_{t+2}\}$ |
| Stage-4* | $\{s_t, a_t, r_t, a_{t+1}\} \rightarrow \{s_{t+1}, s_{t+2}\}$ 
 $\{s_t, a_t, a_{t+1}, r_{t+1}, a_{t+2}, r_{t+2}, r_{t+3}, r_{t+4}, s_{t+5}, a_{t+5}, r_{t+5}, a_{t+6}, a_{t+7}, a_{t+8}, r_{t+8}\} \rightarrow \{a_t, s_{t+1}, s_{t+2}, a_{t+2}, s_{t+3}, s_{t+4}, s_{t+6}, s_{t+9}\}$ 
 $\{s_t, a_t, a_{t+1}, a_{t+2}, r_{t+2}, r_{t+3}, a_{t+4}, r_{t+4}, a_{t+5}, r_{t+5}, a_{t+6}, a_{t+7}, a_{t+8}, r_{t+8}\} \rightarrow \{a_t, s_{t+1}, s_{t+2}, a_{t+2}, s_{t+3}, s_{t+4}, s_{t+6}, a_{t+8}, s_{t+9}\}$ 
 †$\{s_t, a_t, a_{t+1}, s_{t+2}, a_{t+2}, a_{t+3}, r_{t+3}, a_{t+4}, r_{t+4}, a_{t+5}, a_{t+7}, s_{t+8}, a_{t+8}, r_{t+8}\} \rightarrow \{s_{t+1}, s_{t+3}, a_{t+4}, s_{t+6}, s_{t+9}\}$ 
 $\{s_t, a_t, a_{t+1}, a_{t+2}, r_{t+3}, a_{t+4}, r_{t+4}, s_{t+5}, a_{t+5}, s_{t+7}, a_{t+7}, s_{t+8}, a_{t+8}, r_{t+8}\} \rightarrow \{s_{t+1}, s_{t+2}, s_{t+3}, a_{t+3}, s_{t+4}, s_{t+6}, a_{t+8}, r_{t+8}, s_{t+9}\}$ |
| Stage-5* | $\{s_t, a_t, r_t, a_{t+1}\} \rightarrow \{s_{t+1}, r_{t+1}\}$ 
 $\{s_t, a_t, a_{t+1}, r_{t+1}\} \rightarrow \{s_{t+1}, a_{t+1}, s_{t+2}, a_{t+2}, r_{t+2}, s_{t+3}\}$ 
 $\{s_t, a_t, r_t, a_{t+1}, r_{t+1}\} \rightarrow \{s_{t+1}\}$ 
 $\{s_t, a_t, a_{t+1}, a_{t+2}, r_{t+2}, r_{t+3}, a_{t+4}, r_{t+4}, s_{t+5}, a_{t+5}, r_{t+5}, a_{t+6}, a_{t+7}, a_{t+8}, r_{t+8}\} \rightarrow \{s_{t+1}, s_{t+2}, s_{t+3}, s_{t+4}, s_{t+6}, a_{t+8}, s_{t+9}\}$ 
 $\{s_t, a_t, r_t, a_{t+1}\} \rightarrow \{s_{t+1}, s_{t+2}\}$ |
| Stage-6 | $\{s_t, a_t, a_{t+1}, a_{t+2}, r_{t+2}, a_{t+3}, r_{t+3}, r_{t+4}, s_{t+5}, a_{t+5}, r_{t+5}, a_{t+6}, a_{t+7}, a_{t+8}, r_{t+8}\} \rightarrow \{s_t, a_t, s_{t+1}, s_{t+2}, a_{t+2}, s_{t+3}, a_{t+3}, a_{t+4}, s_{t+5}, s_{t+6}, a_{t+8}, s_{t+9}\}$ 
 $\{s_t, a_t, a_{t+1}\} \rightarrow \{r_t, s_{t+1}, r_{t+1}, s_{t+2}\}$ 
 $\{s_t, a_t, a_{t+1}, a_{t+2}, r_{t+2}, a_{t+3}, r_{t+3}, a_{t+4}, r_{t+4}, s_{t+5}, a_{t+5}, a_{t+6}, a_{t+7}, s_{t+8}, a_{t+8}, r_{t+8}\} \rightarrow \{a_t, s_{t+1}, s_{t+2}, s_{t+3}, s_{t+6}, a_{t+8}, r_{t+8}, s_{t+9}\}$ 
 $\{s_t, a_t, r_t, a_{t+1}\} \rightarrow \{s_{t+1}, s_{t+2}\}$ 
 $\{s_t, a_t, r_t, a_{t+1}\} \rightarrow \{s_{t+1}, s_{t+2}\}$ |
| Stage-7 | $\{s_t, a_t, r_t, a_{t+1}, r_{t+1}\} \rightarrow \{s_{t+1}, s_{t+2}\}$ 
 $\{s_t, a_t, r_t, a_{t+1}, r_{t+1}\} \rightarrow \{s_{t+1}, r_{t+1}, s_{t+2}\}$ 
 $\{s_t, a_t, r_t, a_{t+1}\} \rightarrow \{r_t, s_{t+1}, s_{t+2}\}$ 
 $\{s_t, a_t, a_{t+1}\} \rightarrow \{s_{t+1}, a_{t+1}, s_{t+2}\}$ 
 $\{s_t, a_t, a_{t+1}, a_{t+2}, r_{t+2}, a_{t+3}, r_{t+3}, a_{t+4}, r_{t+4}, s_{t+5}, a_{t+5}, a_{t+6}, a_{t+7}, a_{t+8}, r_{t+8}\} \rightarrow \{s_{t+1}, s_{t+2}, s_{t+3}, a_{t+3}, a_{t+4}, s_{t+6}, s_{t+8}, s_{t+9}\}$ |

*: Used for cross-validation. †: AARL-State.

Table 13: Top-5 candidates of each stage in Hopper-Hop (State) evolution process

| | Hopper-Hop (Raw) |
|---|---|
| Stage-1 | $\{s_t, a_t\} \rightarrow \{r_t, s_{t+1}\}$ 
 $\{a_t, r_t, s_{t+2}, a_{t+2}, r_{t+2}, s_{t+3}, a_{t+3}, r_{t+3}, s_{t+5}, a_{t+5}, r_{t+5}, a_{t+6}, a_{t+7}, r_{t+7}, a_{t+8}, r_{t+8}\} \rightarrow \{s_t, s_{t+1}, a_{t+1}, s_{t+4}, a_{t+4}, s_{t+6}, s_{t+7}, s_{t+8}, s_{t+9}\}$ 
 $\{s_t, a_t, s_{t+2}, a_{t+3}, r_{t+4}, a_{t+5}, r_{t+5}, s_{t+6}, a_{t+6}, r_{t+7}, r_{t+8}\} \rightarrow \{s_t, r_t, s_{t+1}, a_{t+1}, r_{t+1}, s_{t+2}, s_{t+3}, s_{t+4}, a_{t+6}, s_{t+7}, a_{t+7}, r_{t+7}, a_{t+8}, s_{t+9}\}$ 
 $\{s_t, a_t, s_{t+2}, a_{t+3}, r_{t+3}, a_{t+5}\} \rightarrow \{s_t, a_{t+1}, s_{t+2}, s_{t+3}, r_{t+3}, s_{t+4}, r_{t+4}, r_{t+5}, s_{t+6}\}$ 
 $\{s_t, r_t\} \rightarrow \{s_t, r_t, s_{t+1}\}$ |
| Stage-2 | $\{s_t, a_t, s_{t+1}, a_{t+1}\} \rightarrow \{s_{t+1}, s_{t+2}\}$ 
 $\{s_t, a_t, r_t, s_{t+2}, r_{t+2}\} \rightarrow \{r_{t+1}, s_{t+2}, a_{t+2}, s_{t+3}\}$ 
 $\{s_t, a_t, s_{t+1}, a_{t+1}, a_{t+4}, s_{t+5}, a_{t+5}, s_{t+6}, a_{t+6}\} \rightarrow \{s_{t+2}, r_{t+2}, r_{t+3}, r_{t+4}, s_{t+5}, r_{t+5}, s_{t+6}, a_{t+6}, r_{t+6}\}$ 
 $\{r_t, a_{t+1}, r_{t+1}, s_{t+2}, a_{t+2}, s_{t+3}, a_{t+3}, a_{t+4}, r_{t+4}\} \rightarrow \{s_t, a_t, r_t, s_{t+1}, s_{t+2}, s_{t+3}, s_{t+4}, r_{t+4}\}$ 
 $\{s_t, a_{t+1}, s_{t+2}, a_{t+2}, r_{t+2}\} \rightarrow \{s_{t+1}, r_{t+1}, a_{t+2}, s_{t+3}\}$ |
| Stage-3 | $\{s_t, a_t, s_{t+1}, a_{t+1}\} \rightarrow \{s_t, s_{t+2}\}$ 
 $\{s_t\} \rightarrow \{s_t, a_t, r_t, s_{t+1}\}$ 
 $\{s_t, a_t, r_t, s_{t+2}, a_{t+2}, r_{t+2}\} \rightarrow \{s_t, s_{t+1}, r_{t+1}, s_{t+2}, a_{t+2}, r_{t+2}, s_{t+3}\}$ 
 $\{s_t, a_t, a_{t+1}\} \rightarrow \{s_{t+1}, s_{t+2}\}$ 
 $\{s_t, a_t, a_{t+1}\} \rightarrow \{s_t, s_{t+1}, s_{t+2}\}$ |
| Stage-4 | $\{s_t, r_t, s_{t+2}, a_{t+2}, r_{t+2}\} \rightarrow \{r_{t+1}, a_{t+2}, s_{t+3}\}$ 
 $\{s_t, a_t, s_{t+1}, a_{t+1}\} \rightarrow \{s_{t+2}\}$ 
 $\{s_t, r_t, s_{t+1}, a_{t+1}, r_{t+1}, s_{t+2}, a_{t+2}, r_{t+2}\} \rightarrow \{s_t, r_{t+1}, a_{t+2}, s_{t+3}\}$ 
 $\{s_t, a_{t+1}, s_{t+2}, a_{t+2}, r_{t+2}\} \rightarrow \{s_{t+1}, r_{t+1}, a_{t+2}, s_{t+3}\}$ 
 $\{s_t, a_t, s_{t+1}, a_{t+1}\} \rightarrow \{s_{t+1}, s_{t+2}\}$ |
| Stage-5 | $\{s_t, a_t, r_t, a_{t+1}, s_{t+2}, a_{t+2}, r_{t+2}\} \rightarrow \{s_t, r_{t+1}, a_{t+2}, s_{t+3}\}$ 
 $\{s_t, a_{t+1}, r_{t+1}, s_{t+2}, a_{t+2}, r_{t+2}\} \rightarrow \{s_t, r_{t+1}, s_{t+2}, a_{t+2}, s_{t+3}\}$ 
 $\{s_t, a_t, r_{t+1}\} \rightarrow \{s_t, s_{t+1}, s_{t+2}\}$ 
 $\{s_t, r_t, a_{t+1}, s_{t+2}, r_{t+2}\} \rightarrow \{s_{t+1}, a_{t+2}, s_{t+3}\}$ 
 $\{s_t, a_t, s_{t+1}, a_{t+1}\} \rightarrow \{r_{t+1}, s_{t+2}\}$ |
| Stage-6 | $\{s_t, a_t, a_{t+1}\} \rightarrow \{s_t, s_{t+1}, s_{t+2}\}$ 
 $\{s_t, r_t, s_{t+2}, a_{t+2}, r_{t+2}\} \rightarrow \{s_t, s_{t+1}, s_{t+2}, s_{t+3}\}$ 
 $\{s_t, a_t, a_{t+1}\} \rightarrow \{s_t, s_{t+1}, s_{t+2}\}$ 
 $\{s_t, a_t, r_t, s_{t+1}, a_{t+1}, r_{t+1}\} \rightarrow \{r_t, s_{t+1}, s_{t+2}\}$ 
 $\{s_t, a_t, r_t, a_{t+1}, s_{t+2}\} \rightarrow \{s_t, s_{t+1}, r_{t+1}\}$ |

Table 14: Top-5 candidates of each stage in Quadruped-Run (State) evolution process

| | Quadruped-Run (Raw) |
|---|---|
| Stage-1 | $\{a_t, r_t, s_{t+1}, s_{t+2}, a_{t+2}\} \rightarrow \{s_t, a_{t+1}, s_{t+3}\}$ 
 $\{r_t, s_{t+1}, s_{t+3}, r_{t+3}\} \rightarrow \{s_t, a_t, r_t, s_{t+1}, a_{t+1}, r_{t+1}, s_{t+2}, r_{t+2}, s_{t+3}, a_{t+3}, s_{t+4}\}$ 
 $\{a_t, a_{t+1}, r_{t+1}, s_{t+2}, r_{t+2}, s_{t+3}, a_{t+3}, r_{t+3}\} \rightarrow \{s_t, s_{t+1}, a_{t+2}, s_{t+4}\}$ 
 $\{s_t, a_t, r_{t+1}, a_{t+2}, s_{t+3}, a_{t+3}, r_{t+3}, s_{t+4}, s_{t+5}\} \rightarrow \{a_t, a_{t+1}, r_{t+1}, a_{t+2}, r_{t+3}, a_{t+4}\}$ 
 $\{s_t, a_t, r_t, s_{t+1}, a_{t+1}, s_{t+3}\} \rightarrow \{r_{t+1}, r_{t+2}\}$ |
| Stage-2 | $\{a_t, r_t, a_{t+2}, r_{t+2}, s_{t+3}, a_{t+3}, r_{t+3}, a_{t+4}\} \rightarrow \{s_t, s_{t+1}, a_{t+1}, s_{t+2}, s_{t+4}, s_{t+5}\}$ 
 $\{s_t, a_t, a_{t+1}, r_{t+1}, r_{t+2}, a_{t+3}, a_{t+4}, r_{t+4}, a_{t+5}, r_{t+5}, s_{t+6}, r_{t+6}, a_{t+7}, a_{t+8}, r_{t+8}, s_{t+9}\} \rightarrow \{s_{t+1}, s_{t+2}, a_{t+2}, s_{t+3}, s_{t+4}, s_{t+5}, s_{t+7}, s_{t+8}\}$ 
 $\{a_{t+1}, r_{t+1}, s_{t+2}, a_{t+3}, r_{t+3}\} \rightarrow \{s_t, a_t, r_t, a_{t+1}, a_{t+3}, s_{t+4}\}$ 
 $\{a_t, a_{t+1}, s_{t+2}, a_{t+2}, a_{t+3}, a_{t+4}, a_{t+5}, r_{t+5}, a_{t+6}, r_{t+6}, a_{t+7}, s_{t+8}, a_{t+8}\} \rightarrow \{s_t, s_{t+1}, s_{t+3}, s_{t+4}, s_{t+5}, s_{t+6}, s_{t+7}, s_{t+8}, s_{t+9}\}$ 
 $\{s_t, s_{t+1}, a_{t+1}, s_{t+2}, a_{t+2}, s_{t+3}, s_{t+4}\} \rightarrow \{s_t, a_t, s_{t+1}, s_{t+2}, a_{t+2}\}$ |
| Stage-3 | $\{a_t, a_{t+1}, a_{t+3}, r_{t+3}, r_{t+4}, a_{t+5}, a_{t+7}, r_{t+7}, s_{t+8}, a_{t+8}\} \rightarrow \{s_t, s_{t+1}, s_{t+2}, s_{t+3}, s_{t+4}, a_{t+4}, s_{t+5}, a_{t+5}, r_{t+5}, s_{t+6}, a_{t+6}, r_{t+6}, s_{t+7}, s_{t+9}\}$ 
 $\{a_t, a_{t+1}, a_{t+3}, r_{t+3}, a_{t+5}, a_{t+7}, s_{t+8}, a_{t+8}\} \rightarrow \{s_t, a_t, s_{t+1}, a_{t+2}, s_{t+3}, s_{t+4}, a_{t+4}, s_{t+5}, s_{t+6}, a_{t+7}, s_{t+9}\}$ 
 $\{a_t, r_t, r_{t+2}, s_{t+3}, a_{t+3}, r_{t+3}, r_{t+4}\} \rightarrow \{s_t, s_{t+1}, a_{t+1}, s_{t+2}, a_{t+2}, a_{t+3}, s_{t+4}, s_{t+5}\}$ 
 $\{a_t, a_{t+1}, r_{t+3}, a_{t+4}, r_{t+4}, a_{t+5}, a_{t+7}, r_{t+7}, s_{t+8}\} \rightarrow \{s_t, r_t, s_{t+1}, s_{t+3}, s_{t+4}, s_{t+5}, s_{t+6}, a_{t+6}, s_{t+7}, s_{t+8}, s_{t+9}\}$ 
 $\{s_t, a_t, r_t, r_{t+1}, a_{t+2}, r_{t+2}, s_{t+3}, a_{t+3}, r_{t+3}, s_{t+4}, s_{t+5}\} \rightarrow \{a_t, a_{t+1}, r_{t+1}, a_{t+2}, r_{t+3}\}$ |
| Stage-4 | $\{r_t, r_{t+1}, a_{t+2}, r_{t+2}, s_{t+3}, a_{t+3}, r_{t+3}, s_{t+4}, s_{t+5}\} \rightarrow \{a_{t+2}, r_{t+3}, a_{t+4}\}$ 
 $\{s_t, a_t, r_{t+1}, a_{t+2}, r_{t+2}, s_{t+3}, a_{t+3}, r_{t+3}, s_{t+4}, s_{t+5}\} \rightarrow \{a_t, a_{t+1}, a_{t+2}, r_{t+3}, a_{t+4}\}$ 
 $\{a_t, a_{t+1}, a_{t+3}, r_{t+3}, s_{t+4}\} \rightarrow \{s_{t+1}, r_{t+1}, s_{t+2}, a_{t+2}, r_{t+2}, s_{t+3}\}$ 
 $\{s_t, a_t, r_t, r_{t+1}, a_{t+2}, s_{t+3}, a_{t+3}, r_{t+3}, s_{t+4}, s_{t+5}\} \rightarrow \{a_t, a_{t+2}, r_{t+3}, a_{t+4}\}$ 
 $\{s_{t+2}, a_{t+2}, a_{t+3}\} \rightarrow \{s_t, a_t, a_{t+2}, s_{t+3}, a_{t+3}, s_{t+4}\}$ |
| Stage-5 | $\{a_{t+1}, r_{t+1}, s_{t+2}, a_{t+2}, r_{t+2}, a_{t+3}, r_{t+3}\} \rightarrow \{s_t, a_t, r_t, a_{t+1}, r_{t+1}, a_{t+2}, s_{t+3}, a_{t+3}\}$ 
 $\{a_t, r_t, r_{t+1}, a_{t+2}, r_{t+2}, s_{t+3}, a_{t+3}, r_{t+3}, r_{t+4}\} \rightarrow \{s_t, s_{t+1}, a_{t+1}, s_{t+2}, a_{t+2}, a_{t+3}, s_{t+4}, s_{t+5}\}$ 
 $\{a_t, a_{t+1}, a_{t+3}, r_{t+3}, a_{t+4}, r_{t+4}, a_{t+5}, a_{t+7}, r_{t+7}, s_{t+8}, a_{t+8}\} \rightarrow \{s_t, r_t, s_{t+1}, s_{t+2}, s_{t+3}, s_{t+4}, s_{t+5}, a_{t+5}, s_{t+6}, a_{t+6}, r_{t+6}, s_{t+7}, s_{t+8}\}$ 
 $\{a_t, a_{t+1}, s_{t+2}, a_{t+3}, r_{t+3}, a_{t+4}, a_{t+5}, s_{t+6}, a_{t+7}, s_{t+8}, a_{t+8}\} \rightarrow \{s_t, s_{t+1}, s_{t+2}, a_{t+2}, s_{t+3}, s_{t+4}, a_{t+4}, s_{t+5}, s_{t+6}, s_{t+7}, r_{t+7}, r_{t+8}\}$ 
 $\{s_t, a_t, r_t, r_{t+1}, a_{t+2}, s_{t+3}, a_{t+3}, r_{t+3}, s_{t+5}\} \rightarrow \{a_t, a_{t+2}, r_{t+3}, a_{t+4}\}$ |

