# OpenReview forum: "AARL: Automated Auxiliary Loss for Reinforcement Learning"
_ICLR.cc/2022/Conference — ICLR 2022 Submitted_

### Official Review · Reviewer_xAxX · 2021-11-01

**Correctness:** 2
**Technical Novelty And Significance:** 2
**Empirical Novelty And Significance:** 2
**Recommendation:** 3
**Confidence:** 3

**Details Of Ethics Concerns:**

-

**Main Review:**

Pro:
- The paper is well written an easy to follow
- Error measures are present in both the figures and the table report
- The Appendix is well written and has many details


Cons:
- In several parts of the paper, the authors claim that AARL discovers the “optimal auxiliary loss function for RL”. I think this statement is not supported by any mathematical derivation that proves the optimality. Even if we decide to forget about a proper mathematical derivation of optimality there is still the important issue that the authors only considered one domain: the DeepMind control suite. This is clearly not enough to support the broader claims made by the authors, in the abstract, introduction and conclusion.
- Another issue of this approach is that the search space is done on the loss input (I) and the loss operator (f), but the encoder is kept constant. Changing the architecture of the encoder can have a massive impact on the auxiliary loss, especially the one that requires rollouts. For instance, would the conclusion hold if the CURL encoder is substituted with a Transformer? The authors have an ablation on Encoder Architectures in section 3.4, but it’s limited to the state-based case, which is less interesting than the pixel based one. I believe this is an important limitation as it could drastically change the conclusion of the paper.
- Another significant limitation of using just the DeepMind control suite is that the environment is fully observable, at least in its normal implementation. I assume this is the case also for this paper as I don’t see any details in the main paper or in the appendix that go against this assumption. Claiming generality of a method based on conclusions drawn just on a fully observable environment is very dangerous. One known example of this is actually the action prediction auxiliary loss reported in the paper {s_t, s_t+1} -> {a_t}. Indeed, it is well know that this kind off loss works well in fully observable 2D environments, but it could have severe issues in partially observable 3D environments due to the increase state aliasing problem (e.g. Badia et al., 2020).


**Summary Of The Paper:**

The paper introduces an approach to automatically discover optimal auxiliary loss for RL, the method is called (AARL). The authors claim that AARL outperform baselines on both pixel-based and state-based task on the DeepMind Control Suite. Also using their method the author analyse different auxiliary losses to identify common patterns.

**Summary Of The Review:**

The authors combine different AutoML techniques to automatically derive the best auxiliary loss function for RL. Although AARL is an interesting approach, the claim in the paper are not supported by empirical findings.

---

> ### Author Response · Authors · 2021-11-09
> **Author Reply to Reviewer xAxX**
>
> We sincerely thank you for your comprehensive comments on our paper and please find our answers to your questions below.
> ***
> **Q1**: “....This is clearly not enough to support the broader claims made by the authors”
>
> **A1**: In terms of your question on the “optimal auxiliary loss function for RL”, we will rewrite these statements to make them more clear and accurate. We meant to say that under our bi-level optimization scheme, we can automatically find the best-performing auxiliary loss for **a specific RL task**. In the experiment, we apply AARL on both pixel-based and state-based settings which are quite different to each other. We believe AARL can be easily applied to any RL tasks and search for strong-performing auxiliary losses.
> ***
> **Q2**: “....Changing the architecture of the encoder can have a massive impact on the auxiliary loss….”
>
> **A2**: In terms of the architecture of the encoder, this is indeed an interesting future research direction, but might be out of the scope of our current paper. It would of course be ideal if we can search over all possible architectures and even all possible hyperparameters, but it remains unclear how such a search process can be made efficient.
> ***
> **Q3**: “Another significant limitation of using just the DeepMind control suite is that the environment is fully observable….”
>
> **A3**: In terms of the generality of our method, our main point is that our method can be applied to any RL task. In this sense, it is generally applicable. Studying how auxiliary losses work in partially observable environments can certainly be a very interesting future research direction, however it might be out of the scope for our current paper. Related papers[1] have shown auxiliary losses are able to improve RL in the POMDP setting.
>
> [1] Mengjiao Yang and Ofir Nachum. Representation matters: Offline pretraining for sequential decision making. ICML 2021.
>
> Again we would like to emphasize that our paper is already making a number of important contributions. We are the first to propose a novel method for automatic auxiliary loss discovery in RL, and we provided extensive experiments, analysis and insights. We believe this paper would be of interest to the broader community of ICLR.
> ***
> We will revise our paper based on your valuable reviews. The revised version will be soon posted. Please let us know if you have any further comments. We will try our best to address them and improve our paper.

---

> > ### Author Response · Authors · 2021-11-23
> > **Ablation study on the architecture of the encoder**
> >
> > To further address the concerns of the architecture of the encoder from Reviewer xAxX, we conduct an ablation study on the impact of the architecture of the pixel encoder on the performance of AARL agents. The default architecture used in CURL is a 4-layer convolutional encoder. We test AARL with a convolutional encoder of different depths. The result is shown below. The hyperparameters are the same for all different environments and architectures.
> >
> > The result is shown in this [anonymous link](https://anonymous.4open.science/r/AARL-ICLR22-Rebuttal-DCA7/ablation_pixel_encoder-crop.pdf).
> >
> > Note that even though the auxiliary loss is searched with a 4-layer encoder, the 6-layer convolutional encoder is able to perform better in all 3 environments. This proves that the auxiliary loss function of AARL-Pixel is able to improve RL performance with a deeper and more expressive pixel encoder. Moreover, the ranking of RL performance (6-layer > 4-layer > 2-layer) is consistent across three environments. This shows that the auxiliary loss function of AARL-Pixel does not overfit one specific architecture of encoder.
> >
> > Besides the searched auxiliary loss functions, we emphasize that the main contribution of our paper is introducing a principled and universal approach for auxiliary loss design in RL. To the best of our knowledge, we are the first to derive the optimal auxiliary loss function with an automatic process. Our search framework can be easily applied to arbitrary RL tasks to search for the best auxiliary loss function.

---

> > ### Author Response · Authors · 2021-11-23
> > **Ablation study on the POMDP setting**
> >
> > To further address the concerns of the POMDP setting from Reviewer xAxX, we conduct an ablation study on the POMDP setting to see whether AARL is able to perform well in POMDP. We **random mask** 20% of the state dimensions (e.g., 15 dimensions -> 12 dimensions) to form a POMDP environment.
> >
> > The result is shown in this [anonymous link](https://anonymous.4open.science/r/AARL-ICLR22-Rebuttal-DCA7/ablation_pomdp-crop.pdf).
> >
> > Note that AARL-State consistently outperforms CURL and SAC-DenseMLP in the POMDP setting in Hopper-Hop and Cheetah-Run. The experiment results show that AARL is able to improve RL performance in both fully observable and partially observable MDPs.

---

### Official Review · Reviewer_fPxR · 2021-11-02

**Correctness:** 2
**Technical Novelty And Significance:** 2
**Empirical Novelty And Significance:** 1
**Recommendation:** 3
**Confidence:** 4

**Main Review:**

**Strengths**
* This paper is overall clearly written and easy to follow. The method is easy to understand and relatively straightforward. There are enough details to fully reproduce the training.
* To my knowledge, evolving the optimal loss function for auxiliary RL tasks seems to be a novel approach. The central thesis of better representation learning in RL is a problem of good practical value.

**Weaknesses**

My major concern is the weak experimental results. To elaborate:

For pixel-based Deepmind Control Suite, the strongest baseline that the authors compare to is CURL (Laskin et al., 2020). However, this is quite an outdated baseline. The authors *ignore at least 2 recent strong baselines*:

* Image Augmentation Is All You Need: Regularizing Deep Reinforcement Learning from Pixels. Yarats et al. ICLR 2021 Spotlight. https://openreview.net/forum?id=GY6-6sTvGaf. The algorithm is known as "DrQ"
* Reinforcement Learning with Augmented Data. Laskin et al. NeurIPS 2020. Also known as "RAD".

Both papers (neither cited in the paper) are published on top ML conferences before June 2021, so it is fair to request comparison with these prior SoTAs per the ICLR review guideline. Both papers above are about a very simple idea - vanilla reinforcement learning with simple data augmentation can be an exceptionally strong baseline. In fact, if we compare table 1 of the "DrQ" paper (Yarats et al.) with table 3 of this paper, we will see that DrQ *beats AARL-Pixel in 7 out of 12 tasks. For the other 5 tasks, none of the gains of AARL-Pixel is statistically significant.*

This indicates that even **simple RL with image augmentation can outperform the complicated bilevel optimization and auxiliary loss in AARL**, which greatly undermines this paper's contribution.

Furthermore, in section 3.1, Fig. 4, the author does mention "SAC with data augmentation", which appears to use the same scheme as "Reinforcement Learning with Augmented Data. Laskin et al." (RAD). However, SAC + augmentation (blue dashed line) consistently underperforms CURL (blue dotted line) in Fig. 4, which contradicts the results in RAD. In addition, the numerical results indicated by the lines also disagree with Table 1 in the RAD paper. I believe there are factual errors in these plots.

For state-based DMControl experiments, the paper claims in section 3.2 that "there is no data augmentation in the state- based setting." This is not true. Section 5.4 in the RAD paper discusses simple ways to augment low-dimensional states, and shows that they are highly effective in boosting performance.

Minor comment: table 1 has a misnomer. "Inverse dynamics" typically means action inference, instead of predicting the previous state.

**Summary Of The Paper:**

Reinforcement learning agents can benefit from auxiliary tasks. This paper introduces an evolutionary algorithm to automate the design of auxiliary loss functions, based on prior approaches like forward dynamics, inverse dynamics, contrastive state representation learning, etc. The evolution solves a bilevel optimization problem in which the inner loop is regular RL training while the outer loop evolves the loss function. Experiments are conducted on the Deepmind Control Suite for both pixel and state-based observations.

**Summary Of The Review:**

The paper ignores at least 2 simple but important prior works, and do not outperform the results in those baselines. The contribution is greatly undermined by the fact that simple RL with image augmentation can outperform the complicated bilevel optimization and auxiliary loss in AARL. Some experiment plots may contain factual errors.

---

> ### Author Response · Authors · 2021-11-09
> **Author Reply to Reviewer fPxR (1/2)**
>
> We sincerely thank you for pointing out the 2 papers on data-augmentation in RL, we will provide discussion on these papers in our revision. However, we would like to emphasize a few important points here:
> ***
> **Q1**: “....in section 3.1, Fig. 4, the author does mention "SAC with data augmentation", which appears to use the same scheme as "Reinforcement Learning with Augmented Data.....which contradicts the results in RAD.”
>
> **A1**: "SAC with data augmentation" as reported in our paper **is not the same** as RAD. In RAD and DrQ, these papers discussed how different ways of augmenting the data can have different effects on the performance, and they reported results with the best data augmentation scheme they find. In our case, we are treating CURL as our main baseline, we are using the data augmentation scheme in CURL, and not the one in RAD. The reasons for choosing CURL as our main baseline are: 1) CURL is still one of the best auxiliary loss + RL algorithms and has a very simple design; 2) Choosing the same data augmentation scheme as in CURL enables us to conduct a fair comparison to CURL, which helps us conclude that the performance gain of AARL over CURL is brought by a better auxiliary loss searched by AARL and not by other factors.
> ***
> **Q2**: “.... the numerical results indicated by the lines also disagree with Table 1 in the RAD paper.”
>
> **A2**: As reported in our paper (Table 7), the hyperparameters used in evolution are different from that in generalization (e.g., we use a smaller batch size of 128 in evolution to speed up AARL training, and all other baselines like SAC and CURL reported in the evolution figure all use the same batch size of 128). Note that the evolution figure (Figure 4) shows AULC score instead of score at 100k or 500k. Thus **our results in fact do not have a “factual error”**. And since our paper does not focus on the topic of data augmentation, we did not cite these papers initially as they are not closely related to our methodology. But we will discuss more data-efficient RL methods like DrQ and RAD in the revision.
> ***
> **Q3**: “This indicates that even simple RL with image augmentation can outperform the complicated bilevel optimization and auxiliary loss in AARL, which greatly undermines this paper's contribution.”
>
> **A3.1**: Our contributions on finding better auxiliary losses are **orthogonal** to these advances in data augmentation techniques. Contrastive loss methods such as CURL actually rely on data augmentation, thus having better data augmentation schemes will likely enhance our results. But again, we emphasize that our main contribution is proposing a principled and universal approach for auxiliary loss design in RL. Though RAD may have better performance than AARL, we argue that “data augmentation for RL” and “auxiliary loss for RL” are two different tracks of improving RL. More importantly, these two techniques are orthogonal to each other and can be easily combined together with a good choice of data augmentation and auxiliary loss (like CURL leverages random crop as the data augmentation and contrastive loss on states as the auxiliary loss).
>
> **A3.2**: Data augmentation is not the same as auxiliary losses and cannot fully replace auxiliary losses even though some recent papers show they can get strong performance. Note that although RAD and DrQ use data augmentation, they still rely on a rich stream of reward signals to learn useful representations. In the RAD paper, the author himself states that _“Real-world applications of RL might involve performing plenty of interactions (or rollouts) with sparse reward signals, and tasks presented to the agent as image-based goals. In such scenarios, CURL and other representation learning methods are likely to be more important even though current RL benchmarks are primarily about single or multi-task reward optimization….Given these subtle considerations, **we believe that both RAD and representation learning methods like CURL will be useful tools for an RL practitioner in future research encompassing data-efficient and generalizable RL**.”_. We argue that in challenging sparse reward environments where the reward signal is hard to obtain, auxiliary loss can still be used and provide a stable learning signal. Thus auxiliary losses have a unique advantage that cannot be fully replaced by just doing data augmentation. Most importantly, We demonstrate there is still a huge potential in discovering better auxiliary losses (in terms of RL performance improvement and generalization capability) in both pixel-based and state-based environments. We conduct statistical analysis of evolution trials and deliver insights about features of auxiliary losses of RL, which we hope will deepen the understanding of auxiliary losses of RL. We believe this paper would be of interest to the broader community of ICLR.
> ***

---

> > ### Author Response · Authors · 2021-11-09
> > **Author Reply to Reviewer fPxR (2/2)**
> >
> > **Q4**: “For state-based DMControl experiments, the paper claims in section 3.2 that "there is no data augmentation in the state- based setting." This is not true….”
> >
> > **A4**: For your question on the sentence "there is no data augmentation in the state-based setting." We meant to say that in our experiments, we do not use data augmentation for the state-based setting, we will rewrite this part to make it more clear.
> > ***
> > **Q5**: “....table 1 has a misnomer. "Inverse dynamics" typically means action inference, instead of predicting the previous state.”
> >
> > **A5**: Thanks for pointing this out! We will also fix the issues in our revision.
> > ***
> > We will revise our paper based on your valuable reviews. The revised version will be soon posted. Please let us know if you have any further comments. We will try our best to address them and improve our paper.

---

> > ### Comment · Reviewer_fPxR · 2021-11-09
> > **Further experiments are necessary to back up the claims**
> >
> > Thanks for your timely and detailed reply. I maintain my position of dissent:
> >
> > 1. The RAD/DrQ papers use the simple and general technique of image shift and cropping, and show huge performance gain over the complicated contrastive machinery of CURL. I do not see any reason to avoid using the best augmentation technique and stick to the suboptimal augmentation in CURL. Both RAD and DrQ have been published for a long time and random shift/cropping has already been widely adopted in many follow-up works.
> >
> > 2. Both data augmentation and auxiliary loss aim to regularize and improve the learned representation. They are strongly correlated and not completely orthogonal. It is also not necessarily true that their performance gains are _additive_. I am unconvinced about the value of auxiliary loss unless you have experiments that show DrQ/RAD + AARL can significantly outperform DrQ/RAD _alone_ on DMControl. If this is not the case, then why would anyone bother to use the extra machinery of AARL instead of simply adding 5 lines of augmentation code?
> >
> > 3. While I agree with the importance of addressing sparse reward tasks, I also need to note that data augmentation in RAD and DrQ can help those tasks as well. In Fig. 4 of the DrQ paper, it showcases quite a few sparse tasks. I am not convinced that AARL significantly outperforms DrQ/RAD in sparse reward settings, unless there is hard experimental evidence presented.

---

### Official Review · Reviewer_Fvi1 · 2021-11-02

**Correctness:** 3
**Technical Novelty And Significance:** 3
**Empirical Novelty And Significance:** 2
**Recommendation:** 5
**Confidence:** 4

**Main Review:**

The paper shows the following advantages:

(+) A method automatically selecting the auxiliary loss function is introduced in this paper. It avoids the hassle of manually designing and testing auxiliary loss functions, as well as provides a more efficient search method than grid-search when dealing with a huge search space of the auxiliary functions.

(+) The experiments are done with either 5 or 15 seeds, providing reasonably reliable results, although more seeds could improve the accuracy of evaluation even more.

There also remain several concerns:

(-) I am concerned about the accuracy of the first pruning step---randomly sample from the search space, then choose the similarity measure with the best performance which is averaged over 15 random seeds. I agree that this method is more computationally efficient than grid search. However, there exists a trade-off between computational efficiency and the accuracy of searching. There is no guarantee that a good candidate can be sampled in the random sampling step. The sampled mask for loss input may not be the best combination of the chosen operator (similarity measure). Thus, it introduces the risk that the best candidate is never selected and evaluated. It might be better to give an analysis on how large this risk is or the probability this situation happens.

(-) In the experimental result section, I am not convinced by the result of Hopper-Hop and Quadruped-Run. With the chosen auxiliary loss function, there exists a high chance for the agent to perform worse than the SAC-DenseMLP baseline---the averaged value of the violin plot is slightly above the SAC-DenseMLP performance, leaving a lot of auxiliary loss function candidates performing worse than this baseline. This makes me worry that the risk of choosing auxiliary loss seems high.

(-) The paper claims that the chosen auxiliary loss can generalize to unseen tasks. But it is not clear to me where this generalization ability comes from: the auxiliary loss selection method itself, which is a searching process, or the setting of training on multiple different and challenging tasks. Multi-source training itself can prevent overfitting. Therefore, if multi-source training is not part of the framework, it may be worth checking how the different number of training environments affects the generalization ability, especially in the case when there is one training task only.


**Summary Of The Paper:**

This paper introduces a method for searching for the best auxiliary loss function from a huge search space automatically, while the best auxiliary loss function is defined as the one that encourages the agent to get a higher return. The searching space of auxiliary loss functions is finite and discrete. It is a combination of (1) manually designed similarity measures, which is used for encouraging the prediction and target to be similar in the auxiliary task, and (2) the binary mask, which is used for selecting auxiliary loss inputs. In this work, the size of the search space is around $4.6 \times 10^{19}$. The space is pruned firstly by random sampling from the similarity measure space and choosing the one showing the highest averaged performance, then the evolutionary algorithm is applied to select top candidates for loss inputs, which is a method in the literature. The chosen auxiliary losses are empirically shown to be helpful with increasing the learning efficiency and have generalization ability to new tasks.


**Summary Of The Review:**

I consider the weakness to outweigh the strength. This work does provide an interesting idea, which is selecting auxiliary loss functions automatically from a huge search space. The empirical results suggest that the average performance of applying the chosen auxiliary task is higher than baselines. However, the risk of using this method seems high---at the first step of pruning the search space, there remains a high chance that good candidates are never tested, thus the pruning result can be based on the evaluation of bad combinations of operators and inputs. Moreover, the experimental result also suggests that this risk exists.

---

> ### Author Response · Authors · 2021-11-09
> **Author Reply to Reviewer Fvi1**
>
> We sincerely thank you for your comprehensive comments on our paper and please find our answers to your questions below.
> ***
> **Q1**: “I am concerned about the accuracy of the first pruning step….”
>
> **A1**: For your concern on the accuracy of the first pruning step, there is indeed a tradeoff between computational efficiency and the accuracy of searching, which is necessary due to the very large search space. In our paper we actually **already have an ablation on the effect of the pruning step**, shown in Figure 7, end of page 8. The results show that having pruning significantly improves our search efficiency, allowing us to discover good candidates much faster. We will further clarify this in the revision.
> ***
> **Q2**: “....I am not convinced by the result of Hopper-Hop and Quadruped-Run. With the chosen auxiliary loss function, there exists a high chance for the agent to perform worse than the SAC-DenseMLP baseline....This makes me worry that the risk of choosing auxiliary loss seems high.”
>
> **A2**: For your concern on the results of Hopper-Hop and Quadruped-Run, there are indeed a number of candidates that are not having good performance in the population at each stage. However, this is not really a problem because on these figures we are not showing a small number of best candidates, but the entire population. And since for each stage, we generate a large number of new candidates from top-performing (top 25%) candidates of the last stage, it is only natural that some of these candidates in the population will not be strong candidates because of random mutation. We would like to point out that in the end, we **only choose the best candidates** for improving RL, and will not use the rest of the population, thus these bad candidates will not be selected.
> ***
> **Q3**: “....it is not clear to me where this generalization ability comes from….”
>
> **A3**: In the field of AutoRL, searching over a set of training environments[1] is common to enhance generalization ability. In our case we also follow this paradigm. For the experiments where we show the proposed method generalize to unseen tasks, we use cross-validation on 3 searched environments to decide the best candidate, and this process likely provides the generalization ability. Note that in our paper, the number of unseen environments is much greater than the number of training environments. (Pixel: 3 training environments and 9 unseen environments. State: 3 training environments and 15 unseen environments). In the AutoML community, how to generalize to unseen tasks remains an open problem. We will further check how the different number of training environments affects the generalization ability and include it in the revised paper.
>
> [1] Co-Reyes, J. D., Miao, Y., Peng, D., Real, E., Levine, S., Le, Q. V., ... & Faust, A. (2021). Evolving reinforcement learning algorithms. ICLR 2021.
> ***
> We will revise our paper based on your valuable reviews. The revised version will be soon posted. Please let us know if you have any further comments. We will try our best to address them and improve our paper.

---

### Official Review · Reviewer_fH26 · 2021-11-08

**Correctness:** 3
**Technical Novelty And Significance:** 2
**Empirical Novelty And Significance:** 3
**Recommendation:** 6
**Confidence:** 4

**Main Review:**

Advantages:
1. The motivation of automatically searching optimal auxiliary loss is good. Handcraft loss function highly depends on researchers' domain knowledge, and its performance can not be secured. The automatic search strategy can significantly reduce locality and improve the RL agent's performance.
2. The design of auxiliary loss function search space, as well as the evolutionary pipeline, is reasonable. The pruning of search space can greatly help to reduce search complexity.
3. The experiments are extensive and supportive. The critical experimental parameters are provided, and thus there should be no issues with the repeatability of the experiments.

Disadvantages:
1. My major concern is about the efficiency of the proposed method AARL. The overall loss function is a combination of reinforcement learning loss and evolutionary loss, both of which are difficult to optimize. The combination of RL and evolution would be even more difficult to converge. What's more, the efficiency could be very low, as the difficulty of convergence would not be a plus, but be a multiplication of RL and evolution.
2. The combination of RL and evolution could be improved. Equation (1) optimizes the two algorithms by a simple weighted sum of the two loss functions. A more practical solution is to alternatively train the two losses. There are two reasons for doing so: 1). The scalabilities of the RL loss, as well as the auxiliary loss, are different, and it is difficult to confirm the weight \lambda; 2). The required training steps for the two algorithms may differ.
3. The contributions are not significant. The major technical contribution is to design a search space and an evolution strategy to derive an optimal auxiliary loss. And also, there is no theoretical justification for the proposed method.

**Summary Of The Paper:**

In this paper, the authors intend to automatically search for the optimal auxiliary loss. To achieve this goal, the authors propose Automated Auxiliary loss for Reinforcement Learning (AARL), which uses an evolutionary search strategy to explore the very large loss space. The paper conducts extensive experiments on the DeepMind Control Suite and shows that the searched auxiliary losses have significantly improved RL performance in both pixel-based and state-based settings.


**Summary Of The Review:**

This paper has good motivation, a good solution for the problem, and extensive and supportive experiments. The major concerns are low efficiency, convergence difficulty, and non-significant contributions.

---

> ### Author Response · Authors · 2021-11-09
> **Author Reply to Reviewer fH26**
>
> We sincerely thank you for your comprehensive comments on our paper and please find our answers to your questions below.
> ***
> **Q1**: “My major concern is about the efficiency of the proposed method AARL….”
>
> **A1**: For your concern on the efficiency of the search process, if we search in a naive way (for example, always uniformly sample potential candidate) then indeed the computation time would be intractable. That is why in our proposed method we used the genetic algorithm as well as auxiliary loss pruning to improve the efficiency. Genetic algorithms are known to be much more efficient than naive random search, and as shown in our pruning ablation in Figure 7, end of page 8, having pruning further improves our efficiency, allowing us to quickly discover good candidates. We will clarify our writing in Equation (1). There is no evolution loss in our optimization target. RL loss $\mathcal{L}_{RL}$ is actually jointly optimized with auxiliary loss $\mathcal{L}$. Further improving the efficiency of our method is certainly also possible, which we will consider for future work.
> ***
> **Q2**: “....A more practical solution is to alternatively train the two losses….”
>
> **A2**: For your point on alternatively training the two losses, we fully agree with your opinion that such alternative training is more practical, and in fact, **this is exactly what we did in our proposed method**. We will further refine our explanation in the paper to make this point clear and avoid potential misunderstandings.
> ***
> **Q3**: “The contributions are not significant….”
>
> **A3**: For your concern on the contribution of the paper, we would like to point out the fact that such a search procedure on auxiliary losses has **never been tried before** on deep reinforcement learning. Moreover, there are many critical challenges applying AutoML techniques to automatically derive the best auxiliary loss function for RL: 1) We have no prior works to refer to when designing the search space and evolution mutations for auxiliary losses of RL. 2) There is no dataset of millions of images available for RL in the Deepmind Control Suite, so we have to perform optimizing RL and auxiliary losses simultaneously, which makes the searching problem more complex and harder to deal with. We have to make sure the search space contains helpful auxiliary losses and our search strategy could quickly identify them from the large space. As you mentioned, our major technical contribution is to design a search space and an evolution strategy to derive an optimal auxiliary loss. We highlight **more of our major contributions**:
> 1) The proposed approach AARL is principled and universal, and can be easily applied to arbitrary RL tasks to search for strong-performing auxiliary loss functions.
> 2) We demonstrate there is still a huge potential in discovering better auxiliary losses (in terms of RL performance improvement and generalization capability) in both pixel-based and state-based environments. We conduct statistical analysis of evolution trials and deliver insights about features of auxiliary losses of RL, which we hope will deepen the understanding of auxiliary losses of RL. We believe this paper would be of interest to the broader community of ICLR.
> ***
> We will revise our paper based on your valuable reviews. The revised version will be soon posted. Please let us know if you have any further comments. We will try our best to address them and improve our paper.

---

### Public Comment · ~Rishabh_Agarwal2 · 2021-11-11
**Suggestion for improving statistical reliability of results**

Hi authors,

Recently, we found that on DM control (100k and 500k benchmark), there was substantial overlap in 95% CIs of reported mean scores across 6 tasks for recently published methods [1]. Since you have access to 6 tasks with 10 seeds each for 100k/500k, you can report performance metrics which make use of all the 60 seeds for reliable evaluation.

One possible suggestion to build confidence in your results is to report the average probability of improvement over other methods (and other metrics like interquartile mean) with confidence intervals. Performance profiles might also be useful for showing variation in results across different tasks and runs (area under such curve is mean and we can read any percentile from such curves too). The authors can easily do so using the library at https://github.com/google-research/rliable or the [colab](https://bit.ly/statistical_precipice_colab).

Also, note that the individual run scores for DMC 100k/500k benchmark on the 6 tasks (provided by the authors of the corresponding papers) for Dreamer, DrQ, RAD, SAC-AE, PISAC and SLAC are at https://console.cloud.google.com/storage/browser/rl-benchmark-data/dm_control.

[1] Agarwal, R., Schwarzer, M., Castro, P.S., Courville, A. and Bellemare, M.G., 2021. Deep reinforcement learning at the edge of the statistical precipice. In NeurIPS. https://openreview.net/forum?id=uqv8-U4lKBe

---

### Decision · Program_Chairs · 2022-01-20

**Decision:**

Reject

**Comment:**

This paper proposes to use evolutionary methods to learn auxiliary loss functions, demonstrating superior performance vs. typical auxiliary losses previously proposed in the RL literature.

Demonstrating that it is possible to learn auxiliary losses by evolution, both for pixel and state representations, that help train significantly faster (even on new environments) is definitely a meaningful contribution, as acknowledge by the majority of reviewers.

Although many of the original reviews' concerns were addressed by the authors during the discussion period, two major ones were only partially answered, both related to the limited empirical evaluation of the proposed approach (which is crucial for such a contribution that aims to demonstrate an improvement over existing related techniques):
1. The limited set of environments used for evaluation (and in particular the lack of partially observable environments)
2. The fact that the baseline being compared to was CURL, which the paper describes as "the state-of-the-art pixel-based RL algorithm", while reviewers mentioned DrQ and RAD as two more recent (and better) algorithms that were known well ahead of the ICLR submission deadline (note that the more recent DrQ-v2 is now even better). Since the data augmentation techniques used by these algorithms help shape the internal representation, like auxiliary losses do, it would have been important to validate that the proposed technique could be useful when plugged on top of such baselines.

The authors did try their best to address these major concerns during the rebuttal period, but the discussion between reviewers and myself came to the conclusion that this wasn't quite convincing enough yet. I encourage the authors to investigate these points in more depth in a future version of this work so as to make the empirical validation stronger (NB: the links provided in the last comment by authors on Nov. 30th didn't work, but this wasn't the main factor in the decision).